# Igneous meteorites suggest Aluminium-26 heterogeneity in the early Solar Nebula

Evgenii Krestianinov [1] ✉, Yuri Amelin[1,2,3], Qing-Zhu Yin [4], Paige Cary[4], Magdalena H. Huyskens [4], Audrey Miller [4], Supratim Dey[4], Yuki Hibiya [5,6], Haolan Tang[7], Edward D. Young [7], Andreas Pack[8] & Tommaso Di Rocco [8]

The short-lived radionuclide aluminium-26 ($^{26}$Al) isotope is a major heat source for early planetary melting. The aluminium-26 – magnesium-26 ($^{26}$Al-$^{26}$Mg) decay system also serves as a high-resolution relative chronometer. In both cases, however, it is critical to establish whether $^{26}$Al was homogeneously or heterogeneously distributed throughout the solar nebula. Here we report a precise lead-207 – lead-206 ($^{207}$Pb-$^{206}$Pb) isotopic age of 4565.56 ± 0.12 million years (Ma) for the andesitic achondrite Erg Chech 002. Our analysis, in conjunction with published $^{26}$Al-$^{26}$Mg data, reveals that the initial $^{26}$Al/$^{27}$Al in the source material of this achondrite was notably higher than in various other well-preserved and precisely dated achondrites. Here we demonstrate that the current data clearly indicate spatial heterogeneity of $^{26}$Al by a factor of 3-4 in the precursor molecular cloud or the protoplanetary disk of the Solar System, likely associated with the late infall of stellar materials with freshly synthesized radionuclides.

The presence of $^{26}$Al (half-life $T_{1/2} = 0.705$ Ma[1]) in the early Solar System, and its possible role as a heat source for planetary and asteroidal melting and early metamorphism, was predicted[2] more than 60 years ago. The discovery[3] of excesses of $^{26}$Mg in Allende meteorite calcium-aluminium-rich inclusions (CAIs) correlated with the Al/Mg ratio gave evidence that $^{26}$Al was indeed present in the early Solar System at the time of CAI formation and decayed in situ. Subsequent studies[4–6] showed ubiquity of $^{26}$Al in CAIs, chondrules and achondrites. The studies of FUN (fractionation and unidentified nuclear effects) CAIs and isotopically anomalous refractory mineral grains in chondrites[4,7,8] pointed to potential heterogeneity in $^{26}$Al distribution, but the homogeneity or heterogeneity of $^{26}$Al abundance was impossible to access from the $^{26}$Al-$^{26}$Mg data alone, without corresponding absolute Pb-Pb or other independent age data to corroborate the $^{26}$Al-$^{26}$Mg data. The age interval between the formation of CAIs and chondrules measured with the Pb-Pb chronometer[9] was found to be broadly consistent

with the time intervals determined with the $^{26}$Al-$^{26}$Mg chronometer[4,5], reinforcing the notion of the uniform initial distribution of $^{26}$Al throughout early Solar System materials at the level of approximately $5\times10^{-5}$ measured in CAIs, referred to as the canonical ratio in cosmochemical studies.

The difference in initial abundance of $^{26}$Al between the CAI formation region and the rest of the protoplanetary disk where chondrules, achondrites and planets formed was proposed[10] on the basis of precisely measured mass-independent Mg isotopic variations in meteoritic materials with low Al/Mg ratios. However, the reliability of this approach was questioned[11,12]. These authors showed that the conclusion about the uniform $^{26}$Al/$^{27}$Al ratio cannot be inferred only from small differences in $^{26}$Mg/$^{24}$Mg because natural $^{26}$Mg/$^{24}$Mg isotopic variations and/or an CAI isochron rotation due to including genetically unrelated amoeboid olivine aggregates (AOAs) can produce apparent initial $^{26}$Mg isotopic heterogeneity. Recent work[13]

[1]Research School of Earth Sciences, Australian National University, Canberra 2601, Australia. [2]Guangzhou Institute of Geochemistry, Chinese Academy of Sciences, Guangzhou, GD 510640, China. [3]Korea Basic Science Institute, Ochang, Cheongwon, Cheongju, Chungbuk 28119, Korea. [4]Department of Earth and Planetary Sciences, University of California, Davis, Davis, CA 95616, USA. [5]Department of General Systems Studies, University of Tokyo, Komaba 3-8-1, Meguro, Tokyo 153-0041, Japan. [6]Research Center for Advanced Science and Technology, University of Tokyo, Komaba 4-6-1, Meguro, Tokyo 153-8904, Japan. [7]Department of Earth, Planetary and Space Sciences, University of California, Los Angeles, Los Angeles, CA 90095, USA. [8]Geochemistry and Isotope Geology Department, Georg-August-Universität Göttingen, Goldschmidtstraße 1, 37077 Göttingen, Germany. ✉e-mail: evgenii.krestianinov@anu.edu.au

highlighted the limitation of approach [10] based on the then sparse Al-Mg data set for AOAs[10].

A more reliable assessment of homogeneity in $^{26}$Al distribution can be achieved by comparing $^{26}$Al-$^{26}$Mg and Pb-Pb ages for a pair of meteorites or their components, where both chronometers date the same event (i.e., rapid crystallisation of the melt), and parent and daughter elements of both chronometers did not migrate. Progress in meteorite chronology achieved in the last 15 years due to increased precision of $^{26}$Al-$^{26}$Mg[14,15] and Pb-Pb[16,17] dating, the discovery of uranium isotope ($^{238}$U/$^{235}$U) variability in meteorites[18], and improving the accuracy of Pb-Pb ages by using measured rather than assumed $^{238}$U/$^{235}$U[19,20] enabled revisiting the distribution patterns of $^{26}$Al and other short-lived radionuclides in the early Solar System[21]. Application of refined dating methods to recently found very old and pristine achondrites – volcanic angrites[15–17,22] and ungrouped achondrites[21,23–27] – yielded a detailed record of magmatism on the asteroids during the first 5 Ma of the Solar System. The age differences between CAIs and achondrites were found to be typically longer when measured with the $^{26}$Al-$^{26}$Mg chronometer than with the Pb-Pb chronometer. This disagreement has been interpreted[22] as indicating about 4 times lower $^{26}$Al/$^{27}$Al ratio in the zone of the solar protoplanetary disk where the angrite parent body formed, compared to the canonical ratio in the formation zone of CAIs. Further enhancements in the sensitivity of Pb-isotopic analyses made it possible to precisely date individual chondrules[28]. Comparison of Pb-isotopic and $^{26}$Al-$^{26}$Mg dates for the same chondrules from LL3.1 chondrite North West Africa (NWA) 5697 and CV3 chondrite Allende[29] also suggested that these two groups of chondrules were formed from precursors with initial $^{26}$Al/$^{27}$Al around 3 times and 11 times lower than the canonical ratio at the time of CAI formation[30].

However, the interpretation of the disagreement between $^{26}$Al-$^{26}$Mg and Pb-isotopic dates as evidence for initial $^{26}$Al/$^{27}$Al heterogeneity has been disputed. The age data for CAIs and chondrules were called into question[31–33]. Among the suggested possible causes of age inaccuracy are biases in Pb-isotopic ages caused by multi-step partial dissolution that was used in chondrule dating[28], age variability in CAIs[19,30,34], and variability of $^{26}$Al/$^{27}$Al ratios in CAIs[35]. These effects, or their combination, could potentially mimic the patterns of heterogeneous $^{26}$Al/$^{27}$Al distribution in the protoplanetary disk at the time of CAI formation.

Better candidates for reference meteorite materials that can be reliably dated with both $^{26}$Al-$^{26}$Mg and Pb-Pb methods (see Supplementary Discussion 1 for discussion of reliability of isotopic dates) can be found among achondrites. For example, volcanic angrites, a group of rapidly cooled, volatile depleted basaltic rocks that are unbrecciated and unshocked[36], are made of mineral assemblages that facilitate precise $^{26}$Al-$^{26}$Mg and Pb-Pb dating. Some of these meteorites are sufficiently large and homogeneous to be studied by multiple methods and by several research groups. Therefore, volcanic angrites are better suited for reliable and verifiable age dating than chondrules and CAIs that are small and heterogeneous. Volcanic angrites studied so far, including D'Orbigny and Sahara 99555 have similar Pb-isotopic ages[16,17,37], $^{238}$U/$^{235}$U ratios[20,38] and $^{26}$Al/$^{27}$Al ratios at the time of crystallisation[15,22,39]. The largest and best-studied volcanic angrite, D'Orbigny, is widely used as a time anchor in early Solar System chronological studies with short-lived radionuclides[20,25,27,40–43]. Combined precise $^{26}$Al-$^{26}$Mg and Pb-isotopic age data with supporting $^{238}$U/$^{235}$U ratios have been also reported for ungrouped achondrites NWA 2976[21], NWA 6704[23,25] and Asuka 881394[27].

The recent discovery of andesitic achondrite Erg Chech 002 (EC 002) and its exceptionally old age according to the $^{26}$Al-$^{26}$Mg systematics[44,45] provides an opportunity to further explore the initial distribution of $^{26}$Al. Combined regression of 19 plagioclase and 11 pyroxene Al-Mg SIMS (secondary-ion mass spectrometry) analyses[44] yielded an isochron corresponding to $^{26}$Al/$^{27}$Al = (5.72 ± 0.07) × 10$^{-6}$. The MC-ICPMS (multi-collector inductively coupled plasma mass

spectrometry) analysis of six pyroxene and plagioclase fractions and one whole rock[45] yielded a steeper isochron corresponding to $^{26}$Al/$^{27}$Al = (8.89 ± 0.09) × 10$^{-6}$. This result was confirmed by independent MC-ICPMS analyses[46] of one pyroxene, two bulk and four plagioclase fractions, which yielded an isochron corresponding to $^{26}$Al/$^{27}$Al = (8.89 ± 0.79) × 10$^{-6}$. The absence of excess dispersion in all three regressions – which include multiple analyses of two minerals with different rates of Mg diffusion[47,48] and cover a wide range of Al/Mg ratios – indicates that the event dated by $^{26}$Al-$^{26}$Mg isochrons was very rapid, i.e., probably crystallisation rather than cessation of diffusion upon cooling. It also suggests that the minerals remained closed to migration of Al and Mg at micron scale[44] ever since. The difference in the slope between the isochrons based on SIMS[44] and MC-ICPMS[45,46] data is probably caused by uncorrected systematic analytical uncertainty in Al/Mg ratios measured in SIMS due to using Ca-rich plagioclase as a standard in the analysis of Na-rich plagioclase. Such analytical biases due to imperfect matrix matching are common in SIMS[49]. If the shallower slope of the SIMS isochron was caused by slow cooling[45], then the SIMS data for plagioclase would depend on the crystal size and SIMS spot location, and would have been widely dispersed below the MC-ICPMS isochron, rather than forming a tight array without excess scatter[44]. The $^{26}$Al/$^{27}$Al ratio yielded by isochrons for EC 002 is ca. 4–5 times higher than the initial $^{26}$Al/$^{27}$Al in the previously known oldest achondrites NWA 11119[42] and Asuka 881394[27].

The chronology of EC 002 also was studied using a short-lived Manganese-53 – Chromium-53 ($^{53}$Mn-$^{53}$Cr) chronometer. Two independent analyses yielded discrepant ages of 4565.56 ± 0.59 Ma[50] and 4566.66 ± 0.56 Ma[51], both anchored to D'Orbigny angrite. The older $^{53}$Mn-$^{53}$Cr age[51] has been explained[50] to be a result of the incorporation of xenolithic material. This age disagreement, however, requires further and thorough petrologic and isotopic investigation of various types of pyroxene megacrysts that are present in EC 002.

In this study, we conduct a precise determination of the Pb-Pb age of EC 002. Our findings, when combined with published $^{26}$Al-$^{26}$Mg data for this meteorite and compared to other achondrites, clearly demonstrate a heterogeneous distribution of $^{26}$Al in the early Solar Nebula.

## Results

### U-Pb and $^{238}$U/$^{235}$U isotopic data

We determined the Pb-isotopic age of EC 002 by analysing 15 pyroxene, 7 whole rock and 1 plagioclase fractions for Pb isotopic composition (analytical procedures, data table and detailed presentation of results are in "Methods" section, Supplementary Data 1 and Supplementary Notes). Five of these fractions (two pyroxenes, two whole rocks and plagioclase) were also analysed for U and Pb concentrations to assess the concordance of the U-Pb system. The isochron, which includes 25 analyses of final leachates and residues, yields a Pb-Pb age of 4565.56 ± 0.12 Ma, MSWD (mean square weighted deviation) = 1.3 (Fig. 1a). Our result is in agreement within uncertainties with the Pb-Pb age[46] of 4565.87 ± 0.30 Ma (three-point isochron with MSWD = 3.3) but is more precise and accurate. The accuracy of our Pb-Pb isochron age is confirmed by the agreement with the weighted average $^{207}$Pb*/$^{206}$Pb* (* denotes radiogenic) model age of 4565.53 ± 0.11 Ma (Fig. 1b), calculated assuming that $^{204}$Pb after blank subtraction is the initial Pb of primordial isotopic composition[52,53]. The age is calculated using $^{238}$U/$^{235}$U = 137.8288 ± 0.0054 (MSWD = 2.1, $n$ = 5) measured in acid-leached whole rock fraction ("Methods" section and Supplementary Table 1). The U isotope composition is also consistent within uncertainties with the $^{238}$U/$^{235}$U ratio of bulk rock fraction[46], confirming our results. The uncertainties of the isochron age and weighted average $^{207}$Pb*/$^{206}$Pb* model age shown in Fig. 1 include uncertainties of the U isotope composition.

The measured $^{238}$U/$^{235}$U of EC002 is higher than in other studied achondrites[20,23,27,30,38,54] and terrestrial rocks, both mafic and felsic[54,55],

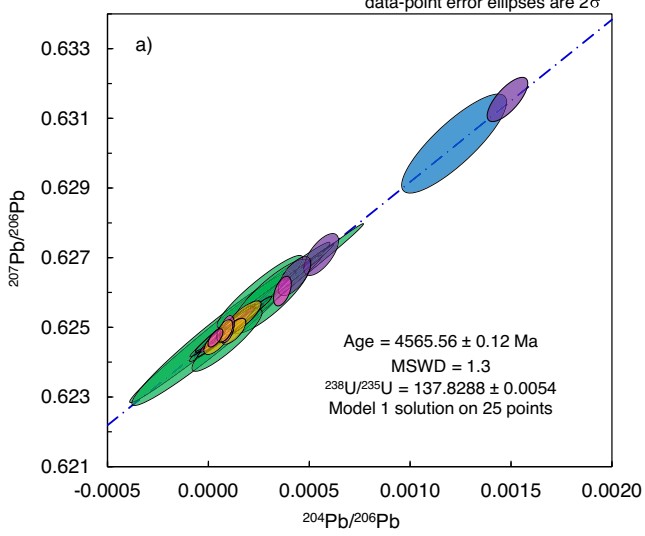

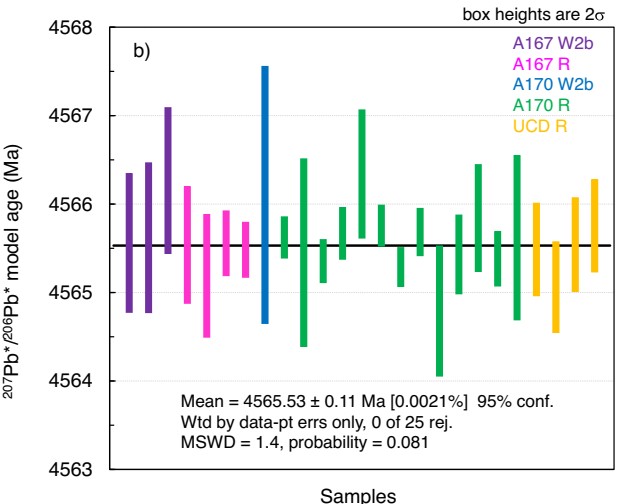

**Fig. 1 | Pb-Pb crystallisation age of Erg Chech 002.** Pb-Pb isochron (dot-dush blue line) plot (**a**) and $^{207}Pb^{*}/^{206}Pb^{*}$ model age (**b**) calculated as the weighted average of each point age used for isochron construction. Both ages are identical within their uncertainties. Source data are provided as a Source Data file. Error ellipses (**a**) and error bars (**b**) are 2σ.

but is similar to the U isotope composition in terrestrial zircon and other uranium-rich accessory minerals[56,57]. It is currently unclear whether the elevated $^{238}U/^{235}U$ in EC 002 is caused by nucleosynthetic heterogeneity, isotope fractionation during melting, or by their combination. These possibilities could be distinguished when meteorites cogenetic to EC 002 are analysed. The difference in $^{238}U/^{235}U$ between EC 002 and other achondrites emphasises the importance of supporting each critical Pb-isotopic date in cosmochemistry by direct measurement of uranium isotopic composition. Using average meteoritic $^{238}U/^{235}U = 137.786 \pm 0.013$ instead of a directly measured value would make the age of EC 002 appear 0.42 Ma younger. This change is 3.5 times larger than the uncertainty of the Pb-isotopic age.

Multiple lines of evidence indicate that the age of $4565.56 \pm 0.12$ Ma derived from the Pb-Pb isochron for EC 002 can be reliably interpreted as the time of its crystallisation. The isochron passes through the point of primordial Pb, thereby eliminating uncertainty in the isotopic composition of initial Pb (Supplementary Fig. 9b). In the U-Pb concordia diagram (Supplementary Fig. 11), the residues and final leachates that were analysed for Pb and U concentrations plot on a Discordia line that passes through zero and has

an upper intercept identical to the Pb-isotopic isochron age, confirming that the U-Pb system remained closed to migration of parent and daughter nuclides from crystallisation till acid leaching in the lab. Furthermore, the U-Pb systems in the residues are nearly concordant (discordance within ±2%, Supplementary Fig. 11 and Supplementary Data 1), demonstrating closed system behaviour even after acid leaching. The reliability of the $^{238}U/^{235}U$ ratio is confirmed by the consistency between the ratios measured on separate dissolutions of the rock with and without acid leaching (Supplementary Fig. 14). The interpretation of the Pb-isotopic age as the time of crystallisation is supported by slower diffusion of Pb in pyroxene compared to Mg diffusion in plagioclase, which indicates that the U-Pb chronometer closed either before or simultaneously with the $^{26}Al$-$^{26}Mg$ chronometer[27], and the latter has been shown above to date crystallisation or very rapid cooling. Reliability and high precision of both U-Pb and $^{26}Al$-$^{26}Mg$ ages make EC 002 a promising candidate for assessment of $^{26}Al$ distribution in the early Solar System.

**Ti-50, Cr-54, Sr-84 and O isotopic composition**
To explore the relationship between the initial $^{26}Al/^{27}Al$, and nucleosynthetic affinity (inner vs. outer Solar System), we determined the isotopic abundances of Titanium-50 ($^{50}Ti$), Chromium-54 ($^{54}Cr$), Strontium-84 ($^{84}Sr$) and Oxygen-17 ($^{17}O$) in the same aliquot of EC 002, thereby assuring internal consistency of the data (results are in "Methods" section and Supplementary Tables 2–4). The average $\varepsilon^{50}Ti = -1.08 \pm 0.30$ (ε unit in geochemistry is a part per 10,000 deviation from the standard value), $\varepsilon^{54}Cr = -0.65 \pm 0.10$, determined from two separate dissolutions, $\varepsilon^{84}Sr = 0.06 \pm 0.24$, and $\Delta'^{17}O = -0.145 \pm 0.009$ (University of California Los-Angeles, UCLA) or $-0.114 \pm 0.023$ (University of Göttingen, UG; defined as a part per 1000 vertical deviation from the terrestrial fractionation line in three oxygen isotope plot).

## Discussion
The initial $^{26}Al/^{27}Al$ ratios for ungrouped achondrites, volcanic angrites and CAIs in comparison with their $^{207}Pb$-$^{206}Pb$ ages are plotted in Fig. 2. The initial $^{26}Al/^{27}Al$ in EC 002[45,46] in combination with our Pb-Pb age is clearly resolved (red band) from other studied achondrites (except NWA 2976, which has large uncertainty of the age) but overlaps with the canonical value in CAIs. Well resolved difference in initial $^{26}Al/^{27}Al$ between EC 002 and volcanic angrites (purple band) persists irrespective of choosing the $^{26}Al/^{27}Al$ value in EC 002 measured by SIMS[44] (orange band) or by MC-ICPMS[45,46]. The difference between the initial $^{26}Al/^{27}Al$ values of EC 002 (we chose more reliable MC-ICPMS result[45] which also was supported by independent analysis[46]) and SAH (Sahara) 99555 angrite[22] is about factor of 4. The difference between the same initial $^{26}Al/^{27}Al$ values of EC 002 and D'Orbigny[22] is about factor of 3. This is reliable evidence that $^{26}Al$ was heterogeneously distributed among the feeding zones of parent asteroids of achondrites. This evidence is independent of any data for CAIs and chondrules and is free from ambiguities concerning their U-Pb and Al-Mg systematics. The $^{26}Al$ heterogeneity can also be shown (Supplementary Discussion 2 and Supplementary Fig. 15) as projections of initial $^{26}Al/^{27}Al$ to any point in time, e.g. either the age of CAIs[30] ($4567.30 \pm 0.16$ Ma) or the age of D'Orbigny ($4563.48 \pm 0.14$ Ma). Supplementary Figure 15 clearly shows that there is a significant gap in the projected initial $^{26}Al/^{27}Al$ between EC 002, on the one hand, and volcanic angrites, on the other. Recently published Pb-Pb and Al-Mg data for EC 002[46] obtained by the MC-ICPMS method are shown in Fig. 2 as a light blue band. The difference in the initial $^{26}Al/^{27}Al$ between formation regions of the parent bodies of EC 002 and angrites is still resolved if we use the Pb-Pb age of[46] for EC 002 instead of the age determined in this study (Fig. 2 and Supplementary Fig. 15). However, concerns about the quality of the Pb-Pb isochron[46], e.g., the inherent low reliability of three-point isochrons, and the presence of two non-radiogenic Pb components indicated by

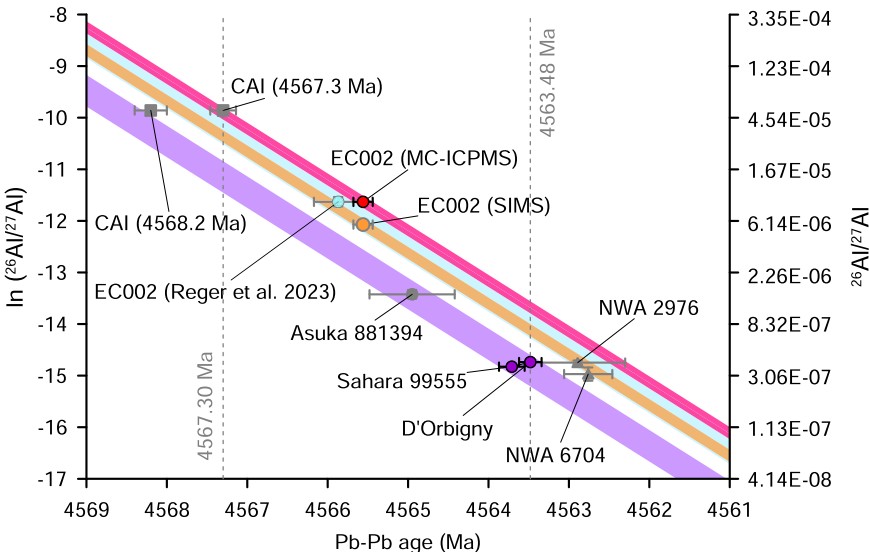

**Fig. 2 | Plot of the $^{26}$Al/$^{27}$Al ratio (natural log scale) versus $^{207}$Pb-$^{206}$Pb ages of achondrites and CAIs.** Each line has a slope of -0.983 corresponding to the decay constant for $^{26}$Al. $^{207}$Pb-$^{206}$Pb data are from[21,23,27,30,34,38,46], $^{26}$Al/$^{27}$Al data are from[10,21,25,27,39,45,46]. Colour bands are shown for EC 002 (red – Pb-Pb age from this study and $^{26}$Al/$^{27}$Al data from ref. [45], light brown – Pb-Pb age from this study and $^{26}$Al/$^{27}$Al from ref. [44], blue – Pb-Pb age and $^{26}$Al/$^{27}$Al data are from[46]) and volcanic angrites (purple). Vertical error bars (2σ) for ln ($^{26}$Al/$^{27}$Al) are smaller than the symbols for all meteorites except NWA 6704 and NWA 2976. Horizontal error bars are 2σ. Source data are provided as a Source Data file.

proximity of the isochron to the composition of the modern terrestrial Pb, suggest the likelihood of uncorrected systematic age uncertainty that would undermine the conclusion about the distribution of $^{26}$Al.

We also plotted the older Pb-Pb age for CAI of 4568.22 ± 0.17 Ma[34] for completeness in Fig. 2 and Supplementary Fig. 15. Our inference about $^{26}$Al/$^{27}$Al heterogeneities are solely based on achondrite observational data, it does not depend on the currently debated specific CAI ages chosen. The ref. [34] age was not corrected for U isotopic composition and, therefore, cannot be considered reliable. Some recent work[58], however, used this data as a justification of the older age of the Solar System than currently accepted[30]. Such an increase in the Solar System age has been invoked to avoid discrepancy between Al-Mg and Pb-Pb chronometers. Al-Mg data[45,46] in combination with our precise Pb-Pb age for EC 002 show that assuming older age of the Solar System does not change the difference between initial $^{26}$Al abundances in EC 002 and volcanic angrites.

Precise initial $\delta^{26}$Mg*$_0$ = −0.009 ± 0.005 derived from the y-intercept of the Al-Mg isochron[45] indicates that the source reservoir of the parent asteroid of EC 002 had separated from the nebula that evolved with solar (=CI chondrite) $^{27}$Al/$^{24}$Mg = 0.101[59] from the initial δ $^{26}$Mg*$_0$ = −0.0379 (Supplementary Fig. 16). As initial $\delta^{26}$Mg*$_0$ of EC 002 intersects the Mg isotopic evolution of the nebula, we can conclude that the time of accretion remains unresolved from the EC 002 crystallisation time (details in Supplementary Discussion and Supplementary Fig. 16, and in ref. [40]). A substantial increase of the Al/Mg ratio occurred during the rapid partial melting of the protolith on the parent body shortly after its accretion.

The ε$^{54}$Cr-Δ$^{'17}$O isotope systematics place EC 002 at the border of HED (howardite-eucrite-diogenite clan), angrites and brachinites regions (Fig. 3d). These data unequivocally point to the non-carbonaceous affinity of EC 002 (Fig. 3), in agreement with its classification as a non-carbonaceous achondrite[44] on the basis of thulium distribution and Cr isotope systematics[50,51]. Both EC002 and volcanic angrites[60,61] are non-carbonaceous achondrites that are thought to have originated in the inner Solar System[62]. The results of this study thus indicate that heterogeneity in $^{26}$Al distribution existed within the inner Solar System.

The integrated patterns of $^{26}$Al/$^{27}$Al, precise ages and stable nucleosynthetic isotope variations that can be established through the

acquisition of extensive and more precise data for both carbonaceous and non-carbonaceous type achondrites, as illustrated in this work, will provide a crucial observational constraints for the models of the collapse of the protosolar molecular cloud, the addition of freshly synthesized nuclides, and asteroid and planet formation in the solar protoplanetary disk.

Our data suggest, analogous with the observed $^{238}$U/$^{235}$U variations[18], that we should exercise caution with using any single value of the initial $^{26}$Al/$^{27}$Al in the studies of meteorite chronology, and in modelling the thermal evolution of asteroids. All relative ages based on Al-Mg system alone must now be reevaluated, including the age differences between chondrules and CAIs. Developing a generalised approach for isotopic dating with Al-Mg and other extinct isotope chronometers that accounts for heterogeneous distribution of the parent radionuclide would allow to produce more accurate and reliable age data for meteorites and asteroidal and planetary materials to advance a better understanding for the formation of our Solar System.

## Methods
All the methods used in this study are destructive, therefore the analysed samples unavailable.

### U-Pb analytical procedures
The U-Pb systematics of EC 002 were studied at the Australian National University (ANU), and at the University of California, Davis (UCD).

Analyses at SPIDE$^2$R lab, Research school of Earth Sciences (RSES), ANU, include two analytical sessions, A167 and A170. A167 mineral fractions were analysed for Pb isotopes and U and Pb concentrations; solutions after U + Pb separation were also used for Sr isotope analysis. Fractions from batch A170 were analysed only for Pb isotopes.

The rock chips were crushed in a boron carbide mortar and sieved into <100 μm and 100–250 μm fractions. The 100–250 μm fraction was washed with ethanol and part of it was separated into mineral fractions by handpicking under a binocular microscope. All mineral or whole-rock fractions were ultrasonically cleaned with ethanol and distilled acetone 2–3 times, 15 min each time. After drying the fractions were weighed and transferred in Teflon vials for step dissolution.

The step dissolution procedure is similar to that described in ref. [40]. The first step is leaching in 0.5 M HNO$_3$ of ambient temperature

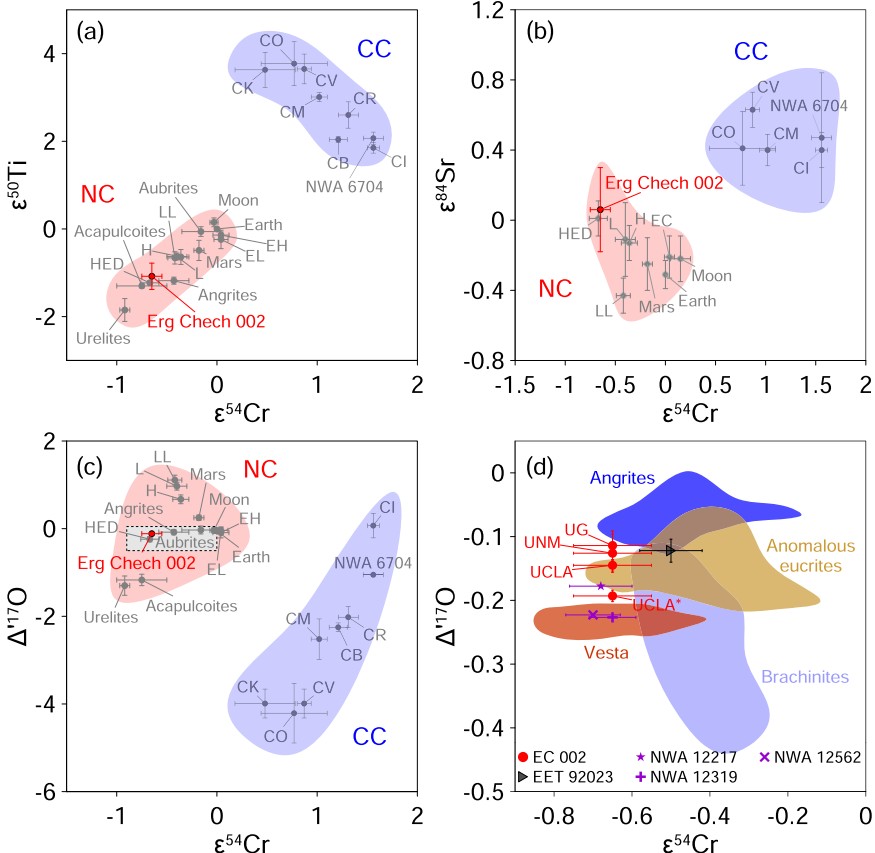

**Fig. 3 | Carbonaceous (CC) and non-carbonaceous (NC) dichotomy among chondrites, achondrites and terrestrial planets inferred from isotopic signatures. a** $\varepsilon^{50}$Ti-$\varepsilon^{54}$Cr, **b** $\varepsilon^{84}$Sr-$\varepsilon^{54}$Cr, **c, d** $\Delta'^{17}$O-$\varepsilon^{54}$Cr. The shaded area in **c** is shown in detail in **d** with data for Vesta, angrites, brachinites, some anomalous eucrites and ungrouped achondrites. Oxygen isotope data for EC 002 are presented in Supplementary Table 4, references for literature data are in Source Data file. All error bars are 2σ.

with ultrasonic agitation (wash 1, W1). The second step includes leaching in hot (110 °C) 7 M $HNO_3$ (W2a) and hot 6 M HCl (W2b) for 2 h in each acid with ultrasonic agitation for 15 min after each hour. The leached minerals are finally dissolved in a mixture of concentrated HF and $HNO_3$ (residue fraction, R) overnight. All fractions were spiked with a mixed $^{202}$Pb-$^{205}$Pb-$^{233}$U-$^{236}$U spike.

Conversion to a soluble form and chemical separation of Pb and U followed the procedures described in detail in and[23]. Chemical separation of Pb has been done using Eichrom AG1x8 anion exchange resin. All solutions passed through columns containing 0.05 ml of the resin twice. U was separated after Pb using Eichrom UTEVA resin. All data were corrected for the average procedure blanks measured for each analytical session. The average procedure blanks for the residues were in the range of 1.2–1.6 pg Pb, and for W2b fractions −0.7–0.9 pg Pb.

The Pb samples were loaded on filaments of zone refined Rhenium (Re) with Aldrich silica gel and measured on a modified MAT 261 thermal ionization mass spectrometer in static multi-collector mode using Faraday cups or peak jumping mode using secondary electron multiplier (SEM). The choice of the detector depends on the isotopic composition of a sample. Thus, all R aliquots have a high $^{206}$Pb/$^{204}$Pb ratio up to 10,000–20,000 and were measured with an array of Faraday cups for getting accurate ratios of all isotopes except $^{204}$Pb; less abundant $^{204}$Pb was measured using SEM after lowering filament current or using the three-isotope method measuring only $^{202-204-205}$Pb using SEM.

The reproducibility and accuracy of measurements were monitored using SRM 981 Pb isotopic standard and the EarlyTime (ET1x) standard, a synthetic mixture of enriched Pb and U isotopes that

imitates radiogenic U-Pb system in a sample with the age of 4560 Ma and containing a small quantity of primordial Pb[63]. Because the concentration of Pb that we detected in the batch A167 analyses was higher than expected and therefore higher than in the standards that were run with batch A167, for the analytical session A170 we prepared and measured standard loads that contained six times more Pb than our regular standard loads of 0.3–0.5 ng. Results of the SRM 981 and Early Time runs are shown in Supplementary Fig. 1. The $^{207}$Pb/$^{206}$Pb ratio of regular SRM 981 loads is 0.91450 ± 0.00041 (2SD, $n = 11$) for Faraday cups, 0.91481 ± 0.00047 ($n = 12$) for SEM. For large SRM 981 standards, the $^{207}$Pb/$^{206}$Pb ratio is 0.91459 ± 0.00007 ($n = 9$) for Faraday cups and 0.91523 ± 0.00106 ($n = 9$) for SEM. The most precise $^{207}$Pb/$^{206}$Pb ratios can be obtained from a combination of Faraday and SEM measurements. Such combined measurements yielded the $^{207}$Pb/$^{206}$Pb value of 0.91460 ± 0.00007 ($n = 9$) for the large standard loads. This value is both precise and in perfect agreement with the certified $^{207}$Pb/$^{206}$Pb ratio of 0.91464 ± 0.00033 for the SRM 981. Most of the radiogenic residue fractions of EC 002 were measured by a combination of SEM and Faraday cups detectors.

For regular loads of ET1x, the most radiogenic standard in the EarlyTime series, the $^{207}$Pb/$^{206}$Pb ratio is 0.62356 ± 0.00099 ($n = 13$) for Faraday cups and 0.62389 ± 0.00038 ($n = 13$) for SEM. For large ET1x $^{207}$Pb/$^{206}$Pb ratio is 0.62360 ± 0.00012 ($n = 10$) for Faraday cups and 0.62427 ± 0.00053 ($n = 14$) for SEM. A combination of Faraday and SEM measurements yields $^{207}$Pb/$^{206}$Pb ratio of 0.62366 ± 0.00013 ($n = 10$).

The summary plot (Supplementary Fig. 1) for $^{207}$Pb/$^{206}$Pb ratios (which are crucial for accurate age determination) revealed systematic bias between Faraday cups (or FAR + SEM combined measurements) and SEM measurements, this effect is especially pronounced for large

standards. The shift corresponds to the $^{207}$Pb/$^{206}$Pb model age correction of approximately -1.5 Ma for ET1x standards, therefore all SEM-only measurements (mainly W2b fractions) were corrected for consistency with the FAR + SEM combined measurements, which are in agreement with certified values. Uncertainties of the standards measurements were propagated during data reduction using an AnySpike excel spreadsheet[64].

Lead isotope measurements were conducted at UCD on three pyroxene fractions weighing between 6.43 and 11 mg and one whole-rock fraction weighing 24.13 mg, which were separated from EC 002, rinsed and sonicated three times in distilled acetone. After rinsing, all four fractions underwent three acid-leaching stages—referred to as "washes"—to separate terrestrial Pb contamination from radiogenic and non-radiogenic Pb original to EC 002. The leftover liquid following these washes was collected for Pb isotope analysis. For Wash 1, all four fractions were sonicated in 0.5 N HNO$_3$ five times for 10 min at a time. For Wash 2, the fractions were heated to 110 °C in 6 N HNO$_3$ on a hotplate for 1 h, twice. Wash 3 is identical to Wash 2, but with 6 N HCl in place of 6 N HNO$_3$. The washes were spiked with a $^{202}$Pb-$^{205}$Pb-$^{233}$U-$^{236}$U tracer for Pb isotopic measurements and U concentration calculations, then dried down and re-dissolved in 2.5 N HCl to convert any fluorides remaining in the washes to chlorides. The washes were dried down again and re-dissolved in 0.5 N HBr.

Following the acid-leaching stages, the remaining solid residues for each fraction were spiked with the same $^{202}$Pb-$^{205}$Pb-$^{233}$U-$^{236}$U tracer and completely dissolved in a 2:1 solution of concentrated HF and HNO$_3$, respectively. The dissolved residues were then dried down, re-dissolved in concentrated HNO3, and dried down again. They were then dissolved in 6 N HCl to convert any remaining fluorides in the residues to chlorides, then dried down and re-dissolved in 0.5 N HBr. Following dissolution in 0.5 N HBr, Pb was separated from matrix elements in the washes and residues via anion exchange column chromatography. For Pb isotope analysis, each sample was loaded with silica gel onto a zone-refined Re filament, and measurements were performed on a *Triton Plus* thermal ionization mass spectrometer (TIMS) at UC Davis.

The *Triton Plus* TIMS is equipped with nine Faraday cups and one secondary electron multiplier (SEM) and has a total of ten amplifiers out of which six have a 10$^{13}$ Ω and four a 10$^{11}$ Ω resistor, which can be connected to any of the Faraday cups through a relay matrix. Prior to measurements, amplifier gain and baseline calibrations were performed and the yield between the SEM and Faraday cup was determined. For all samples, at least a few cycles were analysed in peak jumping mode on the SEM. For the larger samples, Pb was also measured with $^{204}$Pb in the axial SEM, while all other isotopes ($^{202}$Pb, $^{205}$Pb, $^{206}$Pb, $^{207}$Pb and $^{208}$Pb) were collected simultaneously in Faraday cups coupled with 10$^{13}$ Ω resistors. The real-time SEM yield compared to the Faraday cups was determined by an additional measurement of $^{205}$Pb in the axial SEM. Procedural blanks were between 1–2 pg for residues fractions.

During the study, the SRM 981 standard was repeatedly analysed to ensure analytical consistency between analytical sessions at ANU and UCD. $^{207}$Pb/$^{206}$Pb ratio of the SRM 981 standards measured at UCD using SEM is 0.91489 ± 0.00032 (2 SD, n = 9) and has shown small but systematic bias relative to the certified value and the combined FAR + SEM measurements at ANU (Supplementary Fig. 1). UCD data also were normalised to ANU FAR + SEM measurements of the SRM 981 standard: ($^{207}$Pb/$^{206}$Pb)$_{norm}$ = ($^{207}$Pb/$^{206}$Pb)$_{sample}$/($^{207}$Pb/$^{206}$Pb)$_{SRM-981}$ × 0.91460, where 0.91460 is $^{207}$Pb/$^{206}$Pb ratio measured at ANU. Uncertainties of the standards measurements were also propagated during data reduction, similar to ANU data.

## Uranium isotope analysis

Our procedure of sample preparation for U isotope analysis (batch A169) is similar to those described in refs. 23 and 65. The sample was

crushed in a boron carbide mortar and pestle. Sieved whole-rock fractions were pre-cleaned by ultrasonic agitation in ethanol, Milli-Q water and distilled acetone 2–3 times. After this, each fraction was divided into two parts: (1) 30–40% for direct dissolution after pre-cleaning (thereafter "bulk" fractions), and (2) 60–70% for two-step acid leaching. The larger fractions were leached twice in 1 mL of 0.5 HNO$_3$ at 120 °C on a hot plate for 30 min with the following ultrasonic agitation for 10 min to dissolve possible phosphates (thereafter "wash" fractions). These two washes were combined (2 mL in total) for subsequent chemical separation. The residue and untreated bulk fractions were then dissolved in a mixture of concentrated HF and HNO$_3$ in the proportion of 3:1 for 2 days at 120 °C on a hot plate. To convert fluorides into soluble salts, all aliquots were evaporated and re-dissolved in 6–7 mL of concentrated HNO$_3$ at 120 °C for 12 days with 30 min ultrasonic agitation each day. During this period samples were evaporated and re-dissolved again in concentric HNO$_3$ four times (every 3 days) to facilitate the dissolution of fluorides. After evaporation, all aliquots were re-dissolved in 5–6 mL of 6 M HCl at 190 °C in Parr bombs. After complete dissolution, the solutions were spiked with IRMM-3636 $^{233-236}$U double spike, evaporated and then again re-dissolved in 6 M HCl for homogenisation. To be sure that spike and samples are fully homogenised, all aliquots were kept on a hot plate at 120 °C for 3 days.

All aliquots were processed through a two-step chemical separation. For the first step, we used Bio-Rad AG-1×8 anion exchange resin (200-400 mesh) to separate U and Fe from the sample matrix. The matrix elements were eluted with 6 M HCl, and U and Fe were eluted with 0.5 M HNO$_3$. To reduce the amount of organics, the pre-filter resin was put at the bottom of each column. In the second step, we processed U + Fe aliquots through UTEVA resin. Iron was eluted with 3 M HNO$_3$, and U was collected in a mixture of 0.02 M HNO$_3$ and 0.2 M HF. For MC-ICPMS analysis, aliquots were evaporated and re-dissolved in 0.5 M HNO$_3$.

The U isotopic composition was determined on a Thermo *Neptune Plus* MC-ICPMS at the ANU, RSES, in static multi-collector mode. The samples were introduced to the plasma with Aridus desolvating nebulizer. Faraday cups used for measuring $^{238}$U and monitoring $^{229}$Th and $^{232}$Th were connected to the amplifiers with 10$^{11}$ Ω resistors, while the cups used for measuring $^{233}$U, $^{235}$U, and $^{236}$U were connected to the amplifiers with 10$^{12}$ Ω resistors. We achieved the signal intensity of around 1 V per 1 ppb of $^{238}$U with a sample solution uptake of 0.1 mL/min, which corresponds to a total ion yield of approximately 1.5%. Measurements quality was monitored by bracketing with the IRMM-184 uranium isotopic standard[66]. The mass spectrometer background was measured before each 3-4 sample measurements and was subtracted from a sample signal. All data were corrected for instrumental fractionation assuming the exponential law of fractionation. The data for secondary standards: SRM 960 (equivalent to CRM-112a), terrestrial basalts BCR-2 and BHVO-2, and EC 002 were adjusted for consistency with the accepted value of $^{238}$U/$^{235}$U = 137.683 ± 0.020 in IRMM-184 standard[66]. Standards were analysed at various intensities ($^{238}$U signal between around 1 and 20 V) and with the various sample to spike ratios to match the concentrations and sample to spike ratios of the unknowns and confirm the consistency of the data obtained under varying analytical conditions. All EC 002 fractions were analysed at the intensity of 15-17 V.

Measurements of the IRMM-184 isotopic standard with standard to spike ratios of 3, 10 and 30 revealed a linear correlation between $^{238}$U/$^{235}$U and $^{236}$U/$^{238}$U ratios (Supplementary Fig. 12a). To resolve the potential reason for the correlation, we measured the isotopic composition ($^{235}$U/$^{236}$U and $^{238}$U/$^{236}$U ratios) of the IRMM-3636 spike using solutions with concentrations of 1, 3, 10 and 20 ppb (Supplementary Fig. 13). Y-axis intercepts yielded a $^{235}$U/$^{236}$U ratio of 4.8 × 10$^{-5}$ and a $^{238}$U/$^{236}$U ratio of 2.4 × 10$^{-4}$. Usage of these measured ratios $^{236}$U/$^{235}$U = 20,674 and $^{238}$U/$^{235}$U = 4.8873 instead of certified values of

21,988 and 5.1629, respectively, eliminated correlation between $^{238}U/^{235}U$ and $^{236}U/^{238}U$ ratios in the different sample/spike mixtures (Supplementary Fig. 12b). All data were processed with the measured isotopic composition of the IRMM-3636 spike. Results of the standards and EC 002 measurements are presented in Supplementary Table 1.

Normalised $^{238}U/^{235}U$ ratios of secondary standards, SRM 960 and terrestrial basalts, agree with published values[55,66], confirming the reliability of the procedure. EC 002 is found to have the heterogeneous isotopic composition of U. The bulk and residue fractions, which are dominated by intrinsic U, have consistent $^{238}U/^{235}U$ ratios (Supplementary Fig. 14), which are higher than the average Solar System and terrestrial basalts values and which we interpret as the isotopic composition of the meteorite before it was exposed to a terrestrial environment. The isotopic composition of the leachate is displaced towards the average terrestrial uranium (Supplementary Fig. 14), and we interpret it as a mixture between intrinsic uranium, and uranium introduced by weathering.

## $^{54}$Cr isotope systematics

Dissolution, purification and Cr isotopic analysis of Erg Chech 002 were conducted at the University of California, Davis (UC Davis). An interior, fusion-crust-free chip of EC 002 was crushed to a powder using an agate mortar and pestle. A 3 mg aliquot of this whole-rock powder was set aside for oxygen isotope analysis. The remaining mass was prepared for dissolution. The whole-rock powder was placed in a polytetrafluoroethylene (PTFE) Parr bomb digestion capsule, along with twice-distilled (ultrapure), concentrated HF and $HNO_3$ in a 3:1 ratio. The PTFE capsule was sealed in a stainless-steel jacket, and heated in an oven at 190 °C for 96 h. This high-pressure, high-temperature environment is necessary to ensure the complete dissolution of highly refractory phases, such as chromite.

Once completely dissolved, Cr was separated from the sample matrix via a three-column separation procedure according to using one column with an anion-exchange resin (Bio-Rad AG1-X8, 200-400 mesh) and two columns with cation-exchange resin (Bio-Rad AG50W-X8, 200-400 mesh) in a PicoTrace class 10–100 clean lab.

Following elemental separation, the isotopic composition of the purified Cr fractions was measured using a Thermo *Triton Plus* thermal ionization mass spectrometer (TIMS). The Cr fractions were loaded onto four outgassed tungsten filaments, with 3 µg sample per filament. The four sample filaments were bracketed with filaments (two before and two after) loaded with an equal amount (3 µg) of NIST SRM 979 Cr isotopic standard.

Each filament analysis consisted of 1200 ratios, with an 8 s integration time for each ratio. The signal intensity for $^{52}$Cr was set to 10 V (± 15%). A detector gain calibration was completed at the start of each filament analysis, and the amplifiers were rotated and baseline-measured after every block of 25 ratios. The instrumental mass fractionation was corrected using an exponential fractionation law and a $^{50}$Cr/$^{52}$Cr ratio of 0.051859[67]. The $^{54}$Cr/$^{52}$Cr ratio is expressed in ε-notation, or parts per 10,000 deviations from the measured NIST SRM 979 standard:

$$\varepsilon^{x}Cr = \left[ \frac{\left(\frac{^{x}Cr}{^{52}Cr}\right)_{EC002}}{\left(\frac{^{x}Cr}{^{52}Cr}\right)_{SRM979}} - 1 \right] \times 10^4 \qquad (1)$$

where $^{x}$Cr is either $^{53}$Cr or $^{54}$Cr.

Cr isotopic composition was measured in two aliquots (A and B), results are presented in Supplementary Table 2.

## $^{84}$Sr isotope systematics

Strontium isotope compositions were measured in the fractions from batches A167 (after U-Pb separation) and A169 (after U separation). All separation and measurement procedures are described in[68]. Sr was separated from aliquots using columns packed with the Eichrom prefilter resin on the bottom and the Eichrom Sr-spec resin on the top. Samples were loaded in 3 M $HNO_3$, the matrix was eluted with three portions of 3 M $HNO_3$, and then Sr was eluted with three portions of 0.02 M $HNO_3$ and evaporated with $H_3PO_4$. Sr isotopic composition was measured on TritonPlus thermal ionization mass-spectrometer at ANU, RSES, using the three-line acquisition scheme. Sr samples were loaded on outgassed Re filaments with $TaF_5$ solution and slowly dried at 0.4 ampere (A) filament current for 15-20 min, then current was increased with a rate of 0.2 A/min up to dull-red glowing, 2.3-2.4 A. Increasing temperature without slow drying causes effervescence of a sample and an unstable signal during measurements. Amplifier gain calibration was performed for each sample after preheating up to 2500 mA. Typically, each analysis consists of two runs, the first one was operated with a heating rate of 4–5 mA/min and was terminated when the signal of $^{88}$Sr reached an intensity of 18-20 V. After that, the second run has been started using the automatic heating procedure of the Triton software to keep intensity between 80 and 120% relative to the initial intensity and lasted until the sample was completely exhausted from the filament. Generally, one measurement consists of 50–70 blocks with 10 cycles in each block. For both runs the baseline was measured for 30 s before each block; lens focusing, and peak centreing were done every five blocks. One integration lasted 8.4 s with an idle time of 3.0 s. The isobaric interference of Rubidium-87 ($^{87}$Rb) was monitored at $^{85}$Rb and was subtracted online during measurements assuming a natural $^{87}$Rb/$^{85}$Rb of 0.386. The $^{84}$Sr/$^{86}$Sr ratios are reported as relative deviations from the SRM 987 Sr standard as ε notation according to the following formula:

$$\varepsilon^{84}Sr = \left[ \frac{\left(\frac{^{84}Sr}{^{86}Sr}\right)_{Sample}}{\left(\frac{^{84}Sr}{^{86}Sr}\right)_{SRM987}} - 1 \right] \times 10^4 \qquad (2)$$

Results of $^{87}$Sr/$^{86}$Sr and $^{84}$Sr/$^{86}$Sr are presented in Supplementary Table 3.

## $^{50}$Ti isotope systematics

A 1.1 g interior chip of EC 002, free from fusion crust, was crushed in an agate mortar. The powdered sample was sieved through 100 µm and 250 µm mesh screens, and the 100–250 µm fraction was analysed for Ti isotopes without mineral separation. A 24 mg aliquot of this fraction was digested with a concentrated HF–$HNO_3$ mixture at 200 °C using a Parr bomb. The digested sample was converted to a soluble form by repeated evaporation with concentrated $HNO_3$ and dissolved in 6 M HCl. The chemical separation of Ti was performed following the procedure described by[69]. First, the sample in 6 M HCl was loaded onto the column packed with Bio-Rad AG1-X8 anion exchange resin (200-400 mesh), in which Ti is eluted while Fe and U are retained by the resin. Second, Ti was separated from matrix elements including Cr and Ca using Eichrom TODGA resin (50–100 µm mesh). Finally, Ti was purified using the AG1-X8 resin, in which the remaining matrix elements were eluted in 4 M HF as well as 0.4 M HCl+1 M HF, followed by Ti elution in 1 M HCl + 2% $H_2O_2$.

The isotopic composition of the purified Ti fraction was measured using a Thermo Fisher Scientific *Neptune Plus* multiple collectors inductively coupled plasma mass spectrometer (MC-ICPMS) at the University of Tokyo. The sample diluted to a concentration of 100 ppb was introduced to the MC-ICPMS using a CETAC Aridus II desolvating nebulizer with a sample uptake rate of approximately 0.15 mL/min. Measurements were performed using a Jet sample cone and an X skimmer cone with high mass resolution, which resulted in $^{48}$Ti signal intensities of approximately $2.5\times10^{-10}$ A. In addition to five Ti isotopes, $^{43}$Ca, $^{51}$V and $^{53}$Cr were monitored to correct for isobaric interferences on Ti isotopes from $^{46}$Ca, $^{48}$Ca, $^{50}$V and $^{50}$Cr. Data were obtained in dynamic mode from 40 cycles, 2 lines/cycle, 8.4 s integration/line and

4 s idle time between lines. Instrumental mass fractionation was corrected with the exponential law by assuming $^{49}Ti/^{47}Ti = 0.749766$[70]. The sample measurement was bracketed by analyses of an Alfa Aesar Ti standard solution. The $^{50}Ti/^{47}Ti$ ratio of the sample is expressed in ε-notation defined as follows:

$$\varepsilon^{50}\text{Ti} = \left[ \frac{\left(\frac{^{50}\text{Ti}}{^{47}\text{Ti}}\right)_{\text{ErgChech 002}}}{\left(\frac{^{50}\text{Ti}}{^{47}\text{Ti}}\right)_{\text{AlfaAesar}}} - 1 \right] \times 10^4 \qquad (3)$$

Analytical uncertainty on the sample $\varepsilon^{50}Ti$ combined the internal precision (2 SE) and the reproducibility of the standard analyses (2 SD), added in quadrature. For EC 002 we obtained $\varepsilon^{50}Ti$ value of -1.08 ± 0.30.

## Oxygen isotope analysis

The triple oxygen isotope composition is expressed in form of the classical δ-notation with

$$\delta^{17,18}\text{O} = \left( \frac{\left(\frac{^{17,18}\text{O}}{^{16}\text{O}}\right)_{\text{Sample}}}{\left(\frac{^{17,18}\text{O}}{^{16}\text{O}}\right)_{\text{SMOW}}} - 1 \right) * 1000 \qquad (4)$$

where SMOW – standard mean ocean water.

The oxygen isotope anomaly is expressed in the $\triangle'^{17}O$ notation with:

$$\triangle'^{17}\text{O} = 1000 \times \ln\left(\frac{\delta^{17}\text{O}+1}{1000}\right) - 0.528 \times 1000 \times \ln\left(\frac{\delta^{18}\text{O}+1}{1000}\right) \qquad (5)$$

## Oxygen isotopic analysis at the University of Göttingen

The oxygen isotope composition of Erg Chech 002 was determined at the University of Göttingen by infrared (IR) laser fluorination[71]. An aliquot of ~2 mg was loaded into a small, 2-pit sample holder together with San Carlos olivine and then placed into an air lock system. The all-metal air lock was pumped down to ~3 ×10⁻⁶ mbar and heated using heating tape to about 75 °C for 24 h. The air lock system is attached via a CF40 gate valve (with Kel-F gaskets) to the sample chamber. The empty fluorination chamber was evacuated and heated to about 60 °C for 24 h before being exposed to ~100 mbar BrF$_5$ for about 15 min in order to remove any moisture that adheres to the inner chamber walls after the chamber. The moisture is adsorbed to the walls and window when the sample chamber is opened, as necessary for cleaning or changing the BaF$_2$ window. To minimise contamination, the sample chamber is filled with Ar while being open. After the air lock cooled to room temperature, the sample holder containing Erg Chech 002 and San Carlos olivine was introduced through a gate valve into the fluorination chamber. Samples were then exposed to BrF$_5$ (100 mbar) and fluorinated by scanning the laser beam across the sample pit at increasing laser energy up to 45 W. Following fluorination, sample O$_2$ gas was transferred through cold traps and NaCl (for F$_2$ removal) to a 5 Å molecular sieve trap. From this trap, sample O$_2$ was transferred via He gas stream (10 mL min⁻¹) through a 5 Å molecular sieve packed gas chromatography column (3 m, 1/8", 50 °C) into a second 5 Å molecular sieve trap located in front of a Thermo 253 Plus mass spectrometer. After the evacuation of He from this trap, sample O$_2$ was expanded at 50 °C into the bellows of the mass spectrometer. Samples were measured relative to reference gas that was calibrated using O$_2$ released from San Carlos olivine ($\delta^{18}O = 5.23‰$, $\Delta^{17}O_{0.528} = -0.052‰$[72]). The entire fluorination, gas transfer and measurement procedure were timed using LabVIEW to avoid any user-specific effects. On the basis of replicate analyses of San Carlos olivine, the analytical uncertainty was assumed to be around 0.3‰ for $\delta^{18}O$ and 0.016‰ for $\Delta'^{17}O$ (2 SD).

## Oxygen isotopic analyses at UCLA

A whole-rock oxygen isotopic measurement of Erg Chech 002 was performed at UCLA. The sample was loaded into a 316 L stainless steel chamber for analysis. The sample was heated with an infrared lamp through the ZnSe window to approximately 120 °C while pumping for around 3 h to eliminate surface absorbed water. The background pressure in the sample chamber before fluorination was about 10⁻⁷ bar.

Oxygen in the form of O$_2$ gas was extracted from the sample using laser-heating assisted fluorination. Approximately 90 mbar of doubly-distilled F$_2$ was loaded into the sample chamber as the fluorinating agent[73]. The sample was melted in the presence of F$_2$ with a 20 W CO$_2$ laser pulsed at 10 Hz. After complete fluorination, the product O$_2$ was purified by passing through a heated KBr trap with cold traps at liquid nitrogen temperature located on either side. The KBr serves as a getter for excess F$_2$, releasing bromine that is trapped at liquid nitrogen temperature. The traps also sequester SiF$_4$. The extracted O$_2$ was then collected into the 13X molecular sieve cooled by liquid N$_2$. The gases are released from the 13X mol sieve at 200 °C for 30 min and then transferred into a sample vial by freezing onto a silica gel substrate.

The oxygen isotope ratios were determined on O$_2$ using a high-mass-resolution, double-focusing gas-source mass spectrometer at UCLA (Nu Instruments Panorama 001). The mass resolving power (M/DM) of 40,000 used for these measurements is sufficient to resolve mass interferences (e.g., NF⁺ [74]). The $^{32}O_2^+$, $^{33}O_2^+$ and $^{34}O_2^+$ ion beams were measured using Faraday cups with amplifier resistors of $10^{10}$ Ω, $10^{13}$ Ω and $10^{11}$ Ω, respectively. Analyses were obtained from 20 blocks. Each block comprised 30 cycles of sample/reference gas comparisons, and each cycle consists of 30 s integration. The reference gas was calibrated using O$_2$ purified by gas-chromatography from the air. Accuracy is checked additionally by analysing two geostandards for oxygen isotope ratios, San Carlos olivine (SC olivine) and Gore Mountain Garnet. For this study, the $\Delta'^{17}O$ of Erg Chech 002 was anchored to the composition of San Carlos olivine that was used in Göttingen and by Ziegler et al. at University of New Mexico (UNM)[75]. Results are presented in Supplementary Table 4 and compared with the literature data from refs. 76–94 and references therein.

## Data availability

All data generated or analysed during this study are included in this published article and its supplementary information files. Source data for figures are provided with the paper. Source data are provided with this paper.

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

## Acknowledgements
The research was supported by Australian Research Council grant DP190100002 "The history of accretion in our Solar System" (Y.A.).

## Author contributions
The U-Pb chronology was conducted by E.K., Y.A., P.C., M.H.H. and Q.Z.Y. The U isotopic composition and Sr isotope systematics were carried out by E.K. and Y.A. The Cr isotope systematics were performed by S.D., A.M. and Q.Z.Y., the Ti isotope systematics were accomplished by YH. The O isotope data were collected by H.T., E.D.Y., A.P. and T.D.R. The original draft was written by E.K. and Y.A., and all authors contributed to the reviewing and editing of this manuscript.

## Competing interests
The authors declare no competing interests.
