## [Peer Review File · Nature Communications]

REVIEWER COMMENTS

Reviewer #1 (Remarks to the Author):

The question of the heterogeneity of the distribution of aluminum 26 in the early solar system is an extremely important question in cosmochemistry. The authors precisely date the Erg Chech 002 meteorite, which is the oldest known magmatic rock, using U-Pb systematics, and also present new isotopic data that confirm the NC character of this rock. Their data allow them to discuss the fundamental question of ^{26}Al distribution in the solar system 4.5 Ga ago. The data and discussion are of excellent quality, and deserve to be published by Nature Communication. The paper is very well written. I have only minor comments that the authors will have no difficulty incorporating:

-while the authors have perfectly discussed the geochronological aspects of EC 002, they seem to neglect some petrological aspects (which however are not of much importance here). Erg Chech 002 is andesitic, and is probably a very primitive magma (put differently, probably very little different from the magma in equilibrium with its protolith). Therefore, it is unlikely that we will ever have a cogenetic meteorite that corresponds to a more primitive or mafic magma. We could possibly have a more differentiated (cogenetic) rock, or associated cumulates. We could also find other pieces of the same parent body ("mantle", core?, other pieces of the crust?)... The authors should correct lines 151-152 and 201-203 accordingly.

-The compositional ranges given for the different groups of meteorites in figure 3D should be checked. There are of course many isotopic ($\Delta^{17}\text{O}$) compositions of HEDs or angrites given in the literature, but they are not all of equivalent quality. On the other hand, some meteorites have been reclassified (for example Ibitira...). The authors should check the values they use to plot these different groups. The range of $\Delta^{17}\text{O}$ values for HEDs is much narrower than that given in this diagram (see Greenwood et al. 2014 EPSL who discussed earlier values).

-the authors should also consider another recently published dating on Erg Chech (<https://academic.oup.com/mnrasl/article-abstract/515/1/L39/6633663?redirectedFrom=fulltext>).

My comments are very minor. This is an excellent paper.

Reviewer #2 (Remarks to the Author):

Summary: The authors document Pb-Pb dating of the Erg Chech 002, and, combined with Al-Mg dating, come to the conclusion that the short-lived ^{26}Al was heterogeneously distributed in the inner Solar System.

Points to consider: Overall, I think the authors do a very good job at demonstrating the measured Pb-Pb age and I have no reason to doubt the robustness of this age. My main issue with the paper comes from comparisons to other researchers ages and the ramifications of those comparisons.

The authors claim in this manuscript, as have other authors in other published works, that there is ^{26}Al heterogeneity in the Solar System, with the primary evidence that it seems not all Solar System objects line up on a decay line with CAIs and the “canonical” $^{26}\text{Al}/^{27}\text{Al} = 5 \times 10^{-5}$ (essentially Fig. 2 in this manuscript). But it sure seems that EC 002 lines up nicely with CAIs when using the MC-ICPMS data...I am very confused why the authors simply just seem to choose the SIMS data over the plasma data?

As the authors state:

“Well resolved difference in initial $^{26}\text{Al}/^{27}\text{Al}$ between EC 002 and volcanic angrites persists if the value $^{26}\text{Al}/^{27}\text{Al}$ in EC 002 measured by SIMS is accepted instead of the value measured by MC-ICPMS.” Why would you believe SIMS over MCICPMS data!? Did we not learn anything as a community from the SIMS-created ^{60}Fe disaster? SIMS is just simply not well-suited to make these kinds of measurements. It has its uses, but high precision data is not one of them. And it is not like we don't have options...good ICP data exist already. The authors even acknowledge that there are issues with SIMS (~line 116) making this even more puzzling.

It seems that the authors could have just as easily chosen the likely far more reliable MC-ICPMS data and written the paper about homogeneity of ^{26}Al . It could be that I am misunderstanding something crucial in this study, but that quote above from the manuscript is very concerning.

It seems the U-isotopes are slightly different from the rest of the bulk Solar System. I understand that there is some variation around, and I don't doubt this is real, but, as acknowledged by the authors (line 156) the non-standard $^{238}\text{U}/^{235}\text{U}$ represents ~ 0.42 Ma...that is a pretty big difference, so understanding WHY the $^{238}\text{U}/^{235}\text{U}$ is different would be nice. But perhaps this is beyond the scope of this paper, which is understandable.

This would have been unknown to the authors when their paper was submitted, but there was a paper just released in MAPS (Anand et al.) that looks at the Mn-Cr age of EC002 which might be good to discuss in revisions. I of course am not weighing this in my review, but just a note that would be good to discuss if the authors care to.

Fig. 3 typo: a) is $\epsilon^{50}\text{Ti}$, not Sr.

Reviewer #3 (Remarks to the Author):

Review of: U-Pb age of the oldest achondrite points to ^{26}Al heterogeneity in the early Solar System.
By E. Krestianinov et al.

General Comments

Overall, this is an important paper that addresses an important problem. The data are of the highest quality (I would expect no less from the senior authors). I have no serious issues with the conclusions, which seem sound and supported by the data. I do have some minor issues, noted below, but regardless, I recommend publication.

Although well-organized, someone for whom English is a first language needs to go through this paper very carefully. The sentence structure commonly is poor and the wording in places is bad, leading to apparent poor logic. Many sentences are missing one or more articles (“the”, “a”, “an”). I have noted a few egregious examples below, but I don’t have the time or energy to proof-read the entire ms.

In part because of the English issues, the main section of the paper surrounding Figures 2 and 3 is difficult to follow in places. But I think the main problem here is that some of the supporting logic is just left out. I have detailed this below. I do not think my suggestions will change the conclusions of the paper, but the paper will be much easier to read and understand.

Specific Comments

P. 1 Lines 24-26 – “The homogeneity of distribution of radioisotope ^{26}Al in the early solar nebula, a major heat source for early planetary differentiation and foundational assumption to high resolution cosmochronology, remains debatable.”

This sentence is horribly distorted. It needs to be broken into ~ three sentences, something along the lines of:

“The short-lived radionuclide ^{26}Al existed in the early solar nebula and potentially was a major heat source for early planetary melting. The ^{26}Al - ^{26}Mg system also can serve as a high-resolution relative chronometer for early solar system events. In both cases however, it is critical to establish whether ^{26}Al was homogeneously or heterogeneously distributed throughout the solar nebula.”

P. 1 Line 43 – “commonly assumed rather than demonstrated”. This is somewhat unfair and untrue. It implies a certain willful blindness on the part of other authors. Most of the papers that I am aware of on this particular subject (Al-Mg isotopic chronology) have, somewhere, the caveat “assuming nebular homogeneity of ^{26}Al ”. All authors are aware of the problem. And especially for CV3 CAIs, mounting evidence shows that the CAI forming region really was quite homogeneous and so the chronologic interpretation is justified. And the MacPherson – Davis – Zinner 1995 paper certainly never claimed to show perfect nebular homogeneity. As was made very clear, PLACs and FUN CAIs exist, and arguably for the CM accretion region the level of heterogeneity was higher than in the CV accretion region because PLACs are fairly common in CMs. So grain-to-grain heterogeneity existed, but the question is whether the nebula writ large was heterogeneous? That is the question this paper seeks to address, and rightly so.

P. 2 Para. 1 line 53-56 – This is a very debatable point. MacPherson et al. (2012) analyzed an amoeboid olivine aggregate in which the olivine was physically growing on the surface of the CAI. This is not an accidental juxtaposition – the CAI and the olivine are genetically related. So to refer to “genetically unrelated amoeboid olivine aggregates” is at least in one case just wrong. This in turn renders the logic of the entire paragraph questionable. This is one section of the paper that I really disagree with.

P. 2 Para. 2 line 60 – “Crystallization” not “crystallisation”. This entire sentence is poorly structured. From a sentence structure standpoint, it is not clear what the clause “..., and remained closed to migration of parent and daughter elements.” refers to. I think if you add a comma after “components”, and remove the comma after “melt”, it will help.

P. 2 Para. 2 Line 66 – “more pristine chondrites” More pristine than what?

P. 2 Para. 2 Lines 74-78 – Distorted logic. What you mean and what you say are not the same. What you mean is that, at time zero (4.5673 Ga), different parts of the solar nebula had different initial $^{26}\text{Al}/^{27}\text{Al}$ values. No problem. But you are leaving out the “time zero” part. Otherwise, it just sounds like age differences. Please re-write the sentence.

P. 2 Para. 3 Lines 84-86 – Again, you are leaving out the “time zero” part.

P. 3 Para. 2 Line 114 – “remained close to” should be “remained closed to”

P. 3 Para. 3 Line 131 – “yields the Pb-Pb age” should be “yields a Pb-Pb age”

P. 4 Para 1 Line 147 – “but is similar to U isotope composition” should be “is similar to the U isotope composition”

P. 4 Para 2 Line 161 – “in U-Pb concordia” should be “in the U-Pb concordia”

P. 4 Para 1 Line 163 – “and has upper intercept” should be “has an upper intercept”

P. 5 – Figure 2 is the heart of the paper and merits some comment.

First, some minor complaints. All axes need fiducial marks, and they should be darker/longer to render them more visible. To be consistent, either all the fiducial marks should be inside the axes or else outside the axes, not both as shown. My preference is for inside, but it really does not matter. I also strongly suggest that on the right vertical axis, the corresponding non-logarithmic $^{26}\text{Al}/^{27}\text{Al}$ values be shown, so the reader can easily translate the logarithmic values on the left axis to conventional $^{26}\text{Al}/^{27}\text{Al}$ values on the right. It will make the reader’s life much easier.

Second and much more importantly, there must be error bars shown for the $^{26}\text{Al}/^{27}\text{Al}$ values. I doubt that this will in any way affect the conclusions, but the main conclusion of the paper rests on demonstrating that the 9 different $^{26}\text{Al} - \text{Pb-Pb}$ evolution lines are in fact resolved. The reader should not have to seek out Fig. S15 to see the error bars – they need to be on Fig. 2.

Third, no estimate is given in the text about what level of original heterogeneity this implies. By my crude visual estimates, the difference between the uppermost and lowermost evolution lines implies about an 8% level of heterogeneity at 4.5673 Ga. To my recollection, that is about consistent with previous estimates for nebular heterogeneity of ^{26}Al . I think this ought to be mentioned and discussed. All things considered, 8% isn't bad: the inner nebula was surprisingly homogeneous after all.

Fourth, how sensitive is the underlying argument to the precise value of the decay constant for ^{26}Al ? Would shallower/steeper slopes for the evolution lines mean smaller/larger differences in initial $^{26}\text{Al}/^{27}\text{Al}$, or the reverse? I don't know. I also have no idea how well determined the decay constant is, but some discussion ought to be given for what the effect would be.

Fifth, and regarding both Fig. 2 and Figure S15, there is a disconnect regarding the CAI $^{26}\text{Al}/^{27}\text{Al}$ values. Fig. S15 projects the 2 CAIs to a common Pb-Pb age, which yields apparent different initial $^{26}\text{Al}/^{27}\text{Al}$ values. But I seriously question the validity of this. As measured, the two CAIs have identical and equally ancient initial $^{26}\text{Al}/^{27}\text{Al}$ values. Whatever may be the case for isotopic heterogeneity nebula-wide, the CV3 CAIs are remarkably uniform excepting only the very rare FUN CAIs. Also, this exercise requires explicitly validating the greater CAI Pb-Pb age obtained by Bouvier et al. (2010), and as far as I know that value is still subject to significant doubt. Especially in the mind of the second author. What if only the 4.5673 Ga age is accepted for CAIs? Does this affect the Discussion? Probably not, but I think the Discussion ought to at least acknowledge the issue. If I am to believe Fig. S15, either the two CAIs differ in age by 1 m.y. or else formed in dramatically different (isotopically) parts of the solar nebula. There is absolutely no supporting evidence for either possibility other than the Bouvier et al Pb-Pb age. And Fig. 2 and Fig. S15 require the amazing coincidence that two CAIs formed 1 m.y. apart yield identical initial $^{26}\text{Al}/^{27}\text{Al}$ values. Something is wrong, and I am inclined to think it is in one of the two Pb-Pb ages. Your choice which.

P. 5 Para. 2 Line 204 – This statement must be elaborated on in more detail. It comes out of nowhere. Do not force the reader to refer to the Supplemental material.

P. 6 Para. 1 Line 208 – “affinity”. What do you mean? “CC” vs. “NC”?

P. 6 Para. 1 Line 212 – “ $\Delta'17O$ ”. I am assuming that the apostrophe is a typo, but if not then you need to define it. In fact, regardless, you need to define $\Delta17O$ or $\Delta'17O$. There actually is a notation that uses $\Delta'17O$ and it is defined somewhat differently than $\Delta17O$, so be careful here.

P. 6 Para. 1 Line 217 – “ $\Delta17O$ ”. As above, is this really what you mean? Or do you mean “ $\Delta'17O$ ”.

P. 6 Para. 1 Lines 218-223 – “The variation in initial $^{26}Al/^{27}Al$ are thus uncorrelated with formation in ...”. This requires much more elaboration. It is hard to follow. It would really help if CAI values were plotted on Fig. 3. Some (admittedly not all) of the difference between “CC” and “NC” are due solely to the presence of CAIs in the CC. Especially this is true for oxygen, strontium, and titanium.

REVIEWER COMMENTS

Reviewer #1 (Remarks to the Author):

The question of the heterogeneity of the distribution of aluminum 26 in the early solar system is an extremely important question in cosmochemistry. The authors precisely date the Erg Chech 002 meteorite, which is the oldest known magmatic rock, using U-Pb systematics, and also present new isotopic data that confirm the NC character of this rock. Their data allow them to discuss the fundamental question of ^{26}Al distribution in the solar system 4.5 Ga ago. The data and discussion are of excellent quality, and deserve to be published by Nature Communication. The paper is very well written. I have only minor comments that the authors will have no difficulty incorporating:

-while the authors have perfectly discussed the geochronological aspects of EC 002, they seem to neglect some petrological aspects (which however are not of much importance here). Erg Chech 002 is andesitic, and is probably a very primitive magma (put differently, probably very little different from the magma in equilibrium with its protolith). Therefore, it is unlikely that we will ever have a cogenetic meteorite that corresponds to a more primitive or mafic magma. We could possibly have a more differentiated (cogenetic) rock, or associated cumulates. We could also find other pieces of the same parent body ("mantle", core?, other pieces of the crust?)... The authors should correct lines 151-152 and 201-203 accordingly.

We agree with this comment and modified the text accordingly. From lines 151-152, we removed the words "but more mafic". Now the sentence is written as "These possibilities could be distinguished when meteorites cogenetic to EC 002 are analysed", where cogenetic meteorites broadly mean any other pieces of the same parent body. From lines 201-203 we removed the words "and/or fractional crystallizations of the magma" because, indeed, EC 002 is a product of a very primitive magma.

-The compositional ranges given for the different groups of meteorites in figure 3D should be checked. There are of course many isotopic ($\Delta^{17}\text{O}$) compositions of HEDs or angrites given in the literature, but they are not all of equivalent quality. On the other hand, some meteorites have been reclassified (for example Ibitira...). The authors should check the values they use to plot these different groups. The range of $\Delta^{17}\text{O}$ values for HEDs is much narrower than that given in this diagram (see Greenwood et al. 2014 EPSL who discussed earlier values).

Indeed, most of the HEDs form the narrow range of $\Delta^{17}\text{O}$ values. In our dataset, only three samples form outliers from this range. These samples are eucrite Moama with $\Delta^{17}\text{O}$ of -0.195 ± 0.021 (Wiechert et al., 2004, EPSL), howardite NWA 2738 with $\Delta^{17}\text{O}$ of -0.202 ± 0.018 (Greenwood et al., 2017, Chemie der Erde), and howardite Saricicek with $\Delta^{17}\text{O}$ of -0.306 ± 0.029 (Unsalan et al., 2019, M&PS), where the $\Delta^{17}\text{O}$ data were measured by Karen

Ziegler at UNM. These data make the wider range of $\Delta 17O$ values for HEDs. The nature of lower $\Delta 17O$ in Saricicek and other howardite due to mixing with CM chondrites regolith is fully discussed in Unsalan et al. (2019) p.977 and their Fig. 14.

-the authors should also consider another recently published dating on Erg Chech (<https://academic.oup.com/mnrasl/article-abstract/515/1/L39/6633663?redirectedFrom=fulltext>).

We considered this paper, as well as Anand et al., 2022 (M&PS), and Reger et al., 2023 (GCA). The information from these papers is briefly discussed in the text, and the references are added.

My comments are very minor. This is an excellent paper.

Reviewer #2 (Remarks to the Author):

Summary: The authors document Pb-Pb dating of the Erg Chech 002, and, combined with Al-Mg dating, come to the conclusion that the short-lived ^{26}Al was heterogeneously distributed in the inner Solar System.

Points to consider: Overall, I think the authors do a very good job at demonstrating the measured Pb-Pb age and I have no reason to doubt the robustness of this age. My main issue with the paper comes from comparisons to other researchers ages and the ramifications of those comparisons.

The authors claim in this manuscript, as have other authors in other published works, that there is ^{26}Al heterogeneity in the Solar System, with the primary evidence that it seems not all Solar System objects line up on a decay line with CAIs and the “canonical” $^{26}\text{Al}/^{27}\text{Al} = 5 \times 10^{-5}$ (essentially Fig. 2 in this manuscript).

*We think this assessment does not accurately represent our findings. Our study shows that not all Solar System objects line up on a decay line **with each other**, namely there is clear difference in the projected initial $^{26}\text{Al}/^{27}\text{Al}$ between EC 002 and volcanic angrites. The relationship between these projected $^{26}\text{Al}/^{27}\text{Al}$ values do not depend on which CAI values of the $^{26}\text{Al}/^{27}\text{Al}$ and/or $^{207}\text{Pb}/^{206}\text{Pb}$ ages were chosen and the “canonical” CAI value is not relevant, and all the CAI data are shown just for completeness of presentation. If one removes both CAI points in Fig. 2 and Fig. S15, the heterogeneity in $^{26}\text{Al}/^{27}\text{Al}$ among the achondrites (the key focus of this study) are still there. These are primary observational data. Each data point represents two isochrons (Al-Mg fossil isochron slope on Y-axis, and Pb-Pb age on X-axis. None of these depends on CAI and the “canonical” $^{26}\text{Al}/^{27}\text{Al}$).*

But it sure seems that EC 002 lines up nicely with CAIs when using the MC-ICPMS data...

Yes indeed. But the key finding is not whether EC 002 line up with CAI nicely, but EC 002 does not line up with volcanic angrites, no matter whether we use MC-ICPMS or SIMS data. The points being EC002+CAI and volcanic angrites are on different lines (indicating the difference in the initial abundance of ^{26}Al between these reservoirs). This is independent of which CAI absolute ages are chosen. This was not obvious from the data that were available before Al-Mg and Pb-Pb data for EC 002 were compared to similarly precise data for other achondrites.

I am very confused why the authors simply just seem to choose the SIMS data over the plasma data?

As the authors state:

"Well resolved difference in initial $^{26}\text{Al}/^{27}\text{Al}$ between EC 002 and volcanic angrites persists if the value $^{26}\text{Al}/^{27}\text{Al}$ in EC 002 measured by SIMS is accepted instead of the value measured by MC-ICPMS." Why would you believe SIMS over MCICPMS data!? Did we not learn anything as a community from the SIMS-created 60Fe disaster? SIMS is just simply not well-suited to make these kinds of measurements. It has its uses, but high precision data is not one of them. And it is not like we don't have options...good ICP data exist already. The authors even acknowledge that there are issues with SIMS (~line 116) making this even more puzzling.

The reviewer has clearly misunderstood something here. The sentence quoted by the reviewer has no ambiguity. What we stated was that even "IF" the value of $^{26}\text{Al}/^{27}\text{Al}$ in EC 002 measured by SIMS is accepted instead of the preferred value measured by MC-ICP-MS, a well resolved difference in the initial $^{26}\text{Al}/^{27}\text{Al}$ between EC 002 and volcanic angrites is observed. We did not choose SIMS Al-Mg data over MC-ICPMS Al-Mg. It is disappointing that the reviewer obtained an opposite impression, despite the clear statement in our submitted manuscript.

We are well aware of potential accuracy problems in SIMS data likely associated with the sensitivity factor in Al/Mg measurements. The potential pitfall in the SIMS data was fully described in the original submitted manuscript in the lines 115-119. ("The difference in the slope between the isochrons based on SIMS (ref. 44) and MC-ICPMS (ref. 45) data is probably caused by uncorrected systematic analytical uncertainty in Al/Mg ratios measured in SIMS due to using Ca-rich plagioclase as a standard in analysis of Na-rich plagioclase. Such analytical biases due to imperfect matrix matching are common in SIMS (ref. 48)"). What we said was that even if SIMS data is chosen, it does not make EC002 plot on the decay line of ^{26}Al corresponding to the abundance of this nuclide in the source of volcanic angrites. MC-ICPMS that has now yielded identical values in two independent studies is actually more accurate, and we stated this in the text, and support this statement with a brief explanation. What is more important, however, is that the gap in initial $^{26}\text{Al}/^{27}\text{Al}$ between EC 002 and volcanic angrites remains resolved no matter whether we use MC-ICPMS or SIMS data for EC 002.

It seems that the authors could have just as easily chosen the likely far more reliable MC-ICPMS data and written the paper about homogeneity of ^{26}Al . It could be that I am misunderstanding something crucial in this study, but that quote above from the manuscript is very concerning.

Yes, we are afraid that the reviewer indeed misunderstood this aspect of our study. We did not choose SIMS Al-Mg data over MC-ICPMS (as explained above). We think that MC-ICPMS that now has yielded identical values in two independent studies is actually more accurate, and we state this in the text, and support this statement with a brief explanation. And yes, we are well aware of potential accuracy problems in SIMS. What is important, however, is that the gap in initial $^{26}\text{Al}/^{27}\text{Al}$ between EC 002 and volcanic angrites remains resolved no matter whether we use MC-ICPMS or SIMS data for EC 002.

We disagree with the reviewer that by using MC-ICP-MS Al-Mg data would make ^{26}Al homogenous in the early solar system. It indeed eliminates the difference in initial abundance of ^{26}Al between EC 002 and CAIs, but not between EC 002 and volcanic angrites. We also plotted the data for CAI from Bouvier and Wadhwa (2010, NatGeo) paper to clearly show that even increasing the Solar System age does not help to achieve homogeneity in ^{26}Al distribution, as it has been proposed by some researchers (e.g. <https://arxiv.org/abs/2212.00390> or <https://doi.org/10.1016/j.icarus.2023.115427>).

We redrew the Fig. 2 to better show the heterogeneity of ^{26}Al among achondrites and help to avoid misunderstanding. We also changed the wording in the text for better clarity.

It seems the U-isotopes are slightly different from the rest of the bulk Solar System. I understand that there is some variation around, and I don't doubt this is real, but, as acknowledged by the authors (line 156) the non-standard $^{238}\text{U}/^{235}\text{U}$ represents ~0.42 Ma...that is a pretty big difference, so understanding WHY the $^{238}\text{U}/^{235}\text{U}$ is different would be nice. But perhaps this is beyond the scope of this paper, which is understandable.

We mentioned some possibilities like nucleosynthetic heterogeneity or U isotope fractionation during melting. The real understanding of the causes of U isotope variations in meteorites requires dedicated studies, which are beyond the scope of our paper.

This would have been unknown to the authors when their paper was submitted, but there was a paper just released in MAPS (Anand et al.) that looks at the Mn-Cr age of EC002 which might be good to discuss in revisions. I of course am not weighing this in my review, but just a note that would be good to discuss if the authors care to.

Thanks. The literature dedicated to this important meteorite is rapidly growing. We tried to assure that all papers dedicated to EC 002 chronology published so far are mentioned in our revised manuscript.

Fig. 3 typo: a) is $\epsilon^{50}\text{Ti}$, not Sr.

Thanks, done

Reviewer #3 (Remarks to the Author):

Review of: U-Pb age of the oldest achondrite points to ^{26}Al heterogeneity in the early Solar System. By E. Krestianinov et al.

General Comments

Overall, this is an important paper that addresses an important problem. The data are of the highest quality (I would expect no less from the senior authors). I have no serious issues with the conclusions, which seem sound and supported by the data. I do have some minor issues, noted below, but regardless, I recommend publication.

Although well-organized, someone for whom English is a first language needs to go through this paper very carefully. The sentence structure commonly is poor and the wording in places is bad, leading to apparent poor logic. Many sentences are missing one or more articles ("the", "a", "an"). I have noted a few egregious examples below, but I don't have the time or energy to proof-read the entire ms.

In part because of the English issues, the main section of the paper surrounding Figures 2 and 3 is difficult to follow in places. But I think the main problem here is that some of the supporting logic is just left out. I have detailed this below. I do not think my suggestions will change the conclusions of the paper, but the paper will be much easier to read and understand.

We greatly appreciate the reviewer's help with improving this paper.

Specific Comments

P. 1 Lines 24-26 – "The homogeneity of distribution of radioisotope ^{26}Al in the early solar nebula, a major heat source for early planetary differentiation and foundational assumption to high resolution cosmochronology, remains debatable."

This sentence is horribly distorted. It needs to be broken into ~ three sentences, something along the lines of:

"The short-lived radionuclide ^{26}Al existed in the early solar nebula and potentially was a

major heat source for early planetary melting. The ^{26}Al - ^{26}Mg system also can serve as a high-resolution relative chronometer for early solar system events. In both cases however, it is critical to establish whether ^{26}Al was homogeneously or heterogeneously distributed throughout the solar nebula.”

Done. We accept this version with one small change: “serves” rather than “can serve”, because the Al-Mg chronometer is already widely used.

P. 1 Line 43 – “commonly assumed rather than demonstrated”. This is somewhat unfair and untrue. It implies a certain willful blindness on the part of other authors. Most of the papers that I am aware of on this particular subject (Al-Mg isotopic chronology) have, somewhere, the caveat “assuming nebular homogeneity of ^{26}Al ”. All authors are aware of the problem. And especially for CV3 CAIs, mounting evidence shows that the CAI forming region really was quite homogeneous and so the chronologic interpretation is justified. And the MacPherson – Davis – Zinner 1995 paper certainly never claimed to show perfect nebular homogeneity. As was made very clear, PLACs and FUN CAIs exist, and arguably for the CM accretion region the level of heterogeneity was higher than in the CV accretion region because PLACs are fairly common in CMs. So grain-to-grain heterogeneity existed, but the question is whether the nebula writ large was heterogeneous? That is the question this paper seeks to address, and rightly so.

We agree with this comment and thank the reviewer for the clarification. The sentence is rewritten as follows: Subsequent studies ⁴⁻⁶ showed ubiquity of ^{26}Al in CAIs, chondrules and achondrites. The studies of FUN CAIs and isotopically anomalous refractory mineral grains in chondrites ^{4,7,8} pointed to heterogeneity in ^{26}Al distribution at the mineral grain level, but the uniformity of ^{26}Al abundance at a larger scale was hard to access from the ^{26}Al - ^{26}Mg data alone, without corresponding absolute Pb-Pb age data to corroborate the Al-Mg data.

P. 2 Para. 1 line 53-56 – This is a very debatable point. MacPherson et al. (2012) analyzed an amoeboid olivine aggregate in which the olivine was physically growing on the surface of the CAI. This is not an accidental juxtaposition – the CAI and the olivine are genetically related. So to refer to “genetically unrelated amoeboid olivine aggregates” is at least in one case just wrong. This in turn renders the logic of the entire paragraph questionable. This is one section of the paper that I really disagree with.

We agree with the reviewer that this is a debatable point here. We have to note that the genetic relationship between CAIs and AOAs is somewhat ambiguous concept. Their protoliths may have condensed in the same region, or AOAs could have been formed by interaction of CAIs with nebular gas, but this would not guarantee that CAIs and AOAs have the same initial Mg isotope composition. In fact, Krot et al. points out in their study of AOAs (Chemie der Erde (2004) 64, 185–239) that “AOAs and forsterite-rich accretionary rims formed in ^{16}O -rich gaseous reservoirs, probably in the CAI-forming region(s), as aggregates of solar nebular condensates originally composed of forsterite, Fe, Ni-metal, and CAIs” and

“Before and possibly after the aggregation, melilite and spinel in CAIs reacted with SiO and Mg of the solar nebula gas”. The transformation of CAIs to AOAs was therefore not isochemical, and could have added Mg from the gas that had different initial $^{26}\text{Mg}/^{24}\text{Mg}$ than Mg that was initially present in the CAIs. Furthermore, the later studies (Wasserburg et al. (2012) 47, 1980–1997; MacPherson et al. (2017) GCA 201 65–82) have clearly demonstrated variability of initial $^{26}\text{Mg}/^{24}\text{Mg}$ in the formation region of refractory inclusions.

*We re-wrote this part of the text in line with the reviewer’s comment and the above reasoning as follows: **These authors showed that the conclusion about the uniform $^{26}\text{Al}/^{27}\text{Al}$ ratio cannot be inferred only from small differences in $^{26}\text{Mg}/^{24}\text{Mg}$ because natural $^{26}\text{Mg}/^{24}\text{Mg}$ isotopic variations and/or an CAI isochron rotation due uncertain genetic relationship between amoeboid olivine aggregates (AOAs) and CAIs. Furthermore, the studies of Wasserburg et al. (2012) and MacPherson et al. (2017) clearly demonstrated variability of initial $^{26}\text{Mg}/^{24}\text{Mg}$ in the formation region of refractory inclusions. Both of these mechanisms change the intercept of an ^{26}Al - ^{26}Mg isochron and thus mimic initial ^{26}Mg isotopic heterogeneity.***

P. 2 Para. 2 line 60 – “Crystallization” not “crystallisation”. This entire sentence is poorly structured. From a sentence structure standpoint, it is not clear what the clause “..., and remained closed to migration of parent and daughter elements.” refers to. I think if you add a comma after “components”, and remove the comma after “melt)”, it will help.

Thanks. The sentence is modified as follows: A more reliable assessment of homogeneity in ^{26}Al distribution can be achieved by comparing ^{26}Al - ^{26}Mg and Pb-Pb ages for a pair of meteorites or their components, where both chronometers date the same event (i.e., rapid crystallisation of the melt) and parent and daughter elements of both chronometers did not migrate.

P. 2 Para. 2 Line 66 – “more pristine chondrites” More pristine than what?

We removed the word “more”

P. 2 Para. 2 Lines 74-78 – Distorted logic. What you mean and what you say are not the same. What you mean is that, at time zero (4.5673 Ga), different parts of the solar nebula had different initial $^{26}\text{Al}/^{27}\text{Al}$ values. No problem. But you are leaving out the “time zero” part. Otherwise, it just sounds like age differences. Please re-write the sentence.

Done

P. 2 Para. 3 Lines 84-86 – Again, you are leaving out the “time zero” part.

The sentence is changed for clarity. We have to mention, however, that the “time zero” is irrelevant here. The heterogeneity exists at any point in time as long as there live ^{26}Al . Fig. 2

and Fig. S15 illustrates this clearly (e.g. projected to 4567.3 Ma, or 4563.48 Ma, or any other point in time).

P. 3 Para. 2 Line 114 – “remained close to” should be “remained closed to”

Done

P. 3 Para. 3 Line 131 – “yields the Pb-Pb age” should be “yields a Pb-Pb age”

Done

P. 4 Para 1 Line 147 – “but is similar to U isotope composition” should be “is similar to the U isotope composition”

Done

P. 4 Para 2 Line 161 – “in U-Pb concordia” should be “in the U-Pb concordia”

Done

P. 4 Para 1 Line 163 – “and has upper intercept” should be “has an upper intercept”

Done

P. 5 – Figure 2 is the heart of the paper and merits some comment.

First, some minor complaints. All axes need fiducial marks, and they should be darker/longer to render them more visible. To be consistent, either all the fiducial marks should be inside the axes or else outside the axes, not both as shown. My preference is for inside, but it really does not matter. I also strongly suggest that on the right vertical axis, the corresponding non-logarithmic $^{26}\text{Al}/^{27}\text{Al}$ values be shown, so the reader can easily translate the logarithmic values on the left axis to conventional $^{26}\text{Al}/^{27}\text{Al}$ values on the right. It will make the reader's life much easier.

The figure has been redrawn following these suggestions

Second and much more importantly, there must be error bars shown for the $^{26}\text{Al}/^{27}\text{Al}$ values. I doubt that this will in any way affect the conclusions, but the main conclusion of the paper rests on demonstrating that the 9 different ^{26}Al – Pb-Pb evolution lines are in fact resolved. The reader should not have to seek out Fig. S15 to see the error bars – they need to be on Fig. 2.

Done. For clarity, we show error bands only for all EC 002 data and volcanic angrites. Now it is more visible that our Pb-Pb age data provide a good resolution for the conclusion about ^{26}Al heterogeneity. Error bars for $\ln(^{26}\text{Al}/^{27}\text{Al})$ are now included in the plot, but they are

smaller than the symbols size for all meteorites other than NWA 6704 and NWA 2976, and are therefore visible for the latter two meteorites.

Third, no estimate is given in the text about what level of original heterogeneity this implies. By my crude visual estimates, the difference between the uppermost and lowermost evolution lines implies about an 8% level of heterogeneity at 4.5673 Ga. To my recollection, that is about consistent with previous estimates for nebular heterogeneity of ^{26}Al . I think this ought to be mentioned and discussed. All things considered, 8% isn't bad: the inner nebula was surprisingly homogeneous after all.

It is not clear to us how the reviewer got 8% variation here? As illustrated in Fig. 2/ Fig. S15, with a distribution of $^{26}\text{Al}/^{27}\text{Al}$ over a natural log of ~ 1.5 units, which corresponds to a factor of 5-6 in terms of $^{26}\text{Al}/^{27}\text{Al}$ variation (i.e. 5×10^{-5} vs. 1×10^{-5} variation at 4.5673 Ma ago. We explained it in the text.

Fourth, how sensitive is the underlying argument to the precise value of the decay constant for ^{26}Al ? Would shallower/steeper slopes for the evolution lines mean smaller/larger differences in initial $^{26}\text{Al}/^{27}\text{Al}$, or the reverse? I don't know. I also have no idea how well determined the decay constant is, but some discussion ought to be given for what the effect would be.

Please note that on Fig. 2, the Y-axis are experimentally determined observational data of $^{26}\text{Al}/^{27}\text{Al}$ from the fossil isochron of ^{26}Al - ^{26}Mg system; X-axis are absolute age data from Pb-Pb isochron. Both are observational data. The only place where the decay constant comes into play is IF one needs to calculate the ΔT of ^{26}Al age difference between two data points. Nature will have only one slope on this figure (which is the decay constant of ^{26}Al), all slopes are parallel, irrespective of how well the decay constant of ^{26}Al is known. In our study, we used a ^{26}Al half-life of $(7.05 \pm 0.24) \times 10^5$ years determined by Norris et al. (1983). Nishiizumi (2004) reviewed the half-life determinations and calculated an average of three direct measurements (from Samworth et al. (1972), Norris et al. (1983), and Middleton et al. (1983)) of $(7.08 \pm 0.17) \times 10^5$ years, and proposed to use the Norris et al. value. All these three measurements are consistent with each other within a few percent of their uncertainties. Given the very large variation in initial ^{26}Al abundances, circa 1.5 natural log unit of observed heterogeneity, which corresponds to a factor of 5-6 in terms of $^{26}\text{Al}/^{27}\text{Al}$ variation (i.e. 5×10^{-5} vs. 1×10^{-5} variation at 4.5673 Ma ago, equivalent to a factor of 2 half-life of ^{26}Al changes. So, the current ^{26}Al decay constant uncertainties of several percent are trivial compared to the observed variations. The only difference would be the very little change in the slope of the parallel $^{26}\text{Al}/^{27}\text{Al}$ decay curves in Fig. 2. This little change of the slope does not affect the discussion and outcomes of our study. Not even using the most deviant determination of half-life of $(7.8 \pm 0.5) \times 10^5$ by Thomas et al. (1984) with the largest uncertainty would be close to covering the factor of 5 changes in the observed $^{26}\text{Al}/^{27}\text{Al}$ heterogeneity.

Fifth, and regarding both Fig. 2 and Figure S15, there is a disconnect regarding the CAI

$^{26}\text{Al}/^{27}\text{Al}$ values. Fig. S15 projects the 2 CAIs to a common Pb-Pb age, which yields apparent different initial $^{26}\text{Al}/^{27}\text{Al}$ values. But I seriously question the validity of this. As measured, the two CAIs have identical and equally ancient initial $^{26}\text{Al}/^{27}\text{Al}$ values. Whatever may be the case for isotopic heterogeneity nebula-wide, the CV3 CAIs are remarkably uniform excepting only the very rare FUN CAIs. Also, this exercise requires explicitly validating the greater CAI Pb-Pb age obtained by Bouvier et al. (2010), and as far as I know that value is still subject to significant doubt. Especially in the mind of the second author. What if only the 4.5673 Ga age is accepted for CAIs? Does this affect the Discussion? Probably not, but I think the Discussion ought to at least acknowledge the issue. If I am to believe Fig. S15, either the two CAIs differ in age by 1 m.y. or else formed in dramatically different (isotopically) parts of the solar nebula. There is absolutely no supporting evidence for either possibility other than the Bouvier et al Pb-Pb age. And Fig. 2 and Fig. S15 require the amazing coincidence that two CAIs formed 1 m.y. apart yield identical initial $^{26}\text{Al}/^{27}\text{Al}$ values. Something is wrong, and I am inclined to think it is in one of the two Pb-Pb ages. Your choice which.

There are two points that we have to make here, to remove any confusion reflected in the above remark by the reviewer.

First, the main conclusion of our study about the different initial $^{26}\text{Al}/^{27}\text{Al}$ in the formation regions of EC 002 and volcanic angrites is completely independent of any information about CAIs, including their Al-Mg systematics and Pb-isotopic ages. The discussion of validity of the Pb-isotopic and Al-Mg systematics of CAIs is completely outside the scope of this study, and has no influence on our findings. We could have excluded CAIs from the Figures 2 and S15 altogether, and this would have not changed our conclusions at all. We were reluctant to do that only because all previous discussions of the initial ^{26}Al heterogeneity / homogeneity were based on comparison between CAIs and other materials (either chondrules or achondrites), so we wanted to show how CAIs fit in the newly established picture of ^{26}Al heterogeneity.

Second, the points in time to which the initial $^{26}\text{Al}/^{27}\text{Al}$ is projected in Fig. S15 can be arbitrarily chosen. These are just the vertical traverses through the ^{26}Al decay bands shown in Figure 2 with dashed lines. Any discussion of the ^{26}Al heterogeneity is meaningful ONLY if the values were compared at the same time, as the ^{26}Al abundance is changed both by radioactive decay over time and by initial distribution.

We think this comment is quite illuminating. This comment shows that the idea that the CAIs must be at the centre of any discussion of heterogeneous distribution of short-lived radionuclides in the solar protoplanetary disk is so deeply ingrained in the cosmochemical community that even a highly knowledgeable expert as the Reviewer 3 apparently has difficulty looking at the problem from a different angle. We thought that the Figure S15 is intuitive enough, but apparently it is not. We thought that it is clear that CAI data are shown just for illustration, but apparently it is not. Figure S15 is simply a representation of the Fig. 2

in the main text where the vertical dashed lines that were showing the points in time projected in the Figure S15. We hope these explanations help to convey the message of this study correctly.

P. 5 Para. 2 Line 204 – This statement must be elaborated on in more detail. It comes out of nowhere. Do not force the reader to refer to the Supplemental material.

Done, we briefly explained our statement. However, for the full comprehensive understanding the reader needs to refer to supplementary figure S16.

P. 6 Para. 1 Line 208 – “affinity”. What do you mean? “CC” vs. “NC”?

Yes, we added a brief explanation.

P. 6 Para. 1 Line 212 – “ $\Delta'17O$ ”. I am assuming that the apostrophe is a typo, but if not then you need to define it. In fact, regardless, you need to define $\Delta17O$ or $\Delta'17O$. There actually is a notation that uses $\Delta'17O$ and it is defined somewhat differently than $\Delta17O$, so be careful here.

$\Delta'17O$ notation has been defined in the supplementary material. Throughout the text, we used $\Delta'17O$ notation.

P. 6 Para. 1 Line 217 – “ $\Delta17O$ ”. As above, is this really what you mean? Or do you mean “ $\Delta'17O$ ”.

Corrected

P. 6 Para. 1 Lines 218-223 – “The variation in initial $^{26}Al/^{27}Al$ are thus uncorrelated with formation in ...”. This requires much more elaboration. It is hard to follow. It would really help if CAI values were plotted on Fig. 3. Some (admittedly not all) of the difference between “CC” and “NC” are due solely to the presence of CAIs in the CC. Especially this is true for oxygen, strontium, and titanium.

We rewrote and expanded this section.

REVIEWERS' COMMENTS

Reviewer #1 (Remarks to the Author):

I recommend that this paper be accepted. There is just one detail that concerns figure 3d. The three abnormal HEDs should not be considered. It is normal for a howardite that contains trace amounts of CM to be outside the range of pure HEDs. To take into account all the anomalous samples is a nonsense. I suggest, only if other reviewers ask for additional revisions (which I would find excessive), to eliminate the anomalous HEDs in this figure, and to name the field Vesta instead of HED. This is really a minor detail, and this paper should be urgently accepted.

Reviewer #2 (Remarks to the Author):

I am happy with the revised version, and the authors are correct, I did misunderstand some of the topics in the original version regarding the discussion of SIMS v. MC-ICPMS. I think it is well described now, and I think the revised version of Fig 2 is a big improvement. If it were my paper, I would uncomplicate the figure by not showing the SIMS data or Bouvier CAI Pb-Pb age and just discuss why they were not shown, which is completely justified, in my opinion.

My only major issue is that it seems the only disparate values, and the primary cause of the "heterogeneity" argument are those from the Schiller et al. (2015) and a very poorly defined "isochron" for Asuka 881394. There are not a lot of samples that have been done to high fidelity, but it should be pointed out that a single, and largely disputed work, from 2015 is the primary driver of this discrepancy.

I only make this suggestion, and have no further issues that I feel need addressed. I congratulate the authors on a nicely done study.

Reviewer #3 (Remarks to the Author):

I think the authors have done a reasonable job of addressing my and the other reviewers' comments. I recommend publication.

REVIEWERS' COMMENTS

Reviewer #1 (Remarks to the Author):

I recommend that this paper be accepted. There is just one detail that concerns figure 3d. The three abnormal HEDs should not be considered. It is normal for a howardite that contains trace amounts of CM to be outside the range of pure HEDs. To take into account all the anomalous samples is a nonsense. I suggest, only if other reviewers ask for additional revisions (which I would find excessive), to eliminate the anomalous HEDs in this figure, and to name the field Vesta instead of HED. This is really a minor detail, and this paper should be urgently accepted.

Done. We eliminated the three anomalous HEDs (eucrite Moama and two howardites, NWA 2738 and Saricicek) and updated the fig. 3d accordingly.

Reviewer #2 (Remarks to the Author):

I am happy with the revised version, and the authors are correct, I did misunderstand some of the topics in the original version regarding the discussion of SIMS v. MC-ICPMS. I think it is well described now, and I think the revised version of Fig 2 is a big improvement. If it were my paper, I would uncomplicate the figure by not showing the SIMS data or Bouvier CAI Pb-Pb age and just discuss why they were not shown, which is completely justified, in my opinion.

My only major issue is that it seems the only disparate values, and the primary cause of the "heterogeneity" argument are those from the Schiller et al. (2015) and a very poorly defined "isochron" for Asuka 881394. There are not a lot of samples that have been done to high fidelity, but it should be pointed out that a single, and largely disputed work, from 2015 is the primary driver of this discrepancy.

I only make this suggestion, and have no further issues that I feel need addressed. I congratulate the authors on a nicely done study.

In the context of our study, the heterogeneity of ^{26}Al distribution is established primarily on the basis of the difference in initial abundance between EC002 and volcanic angrites. We think that this difference is well established. To compare the initial distribution of ^{26}Al between EC002 and volcanic angrites, we used the ^{26}Al - ^{26}Mg isochrons for volcanic angrites from Schiller et al. (2015), along with earlier studies by Schiller et al. (2010) and Spivak-Birndorf et al. (2009). The ^{26}Al - ^{26}Mg isochrons from all three studies are consistent with each other, except for the data

for SAH 99555, which differs between Schiller et al. (2010) and Schiller et al. (2015). We chose to use the latter data in our analysis, but our conclusions are not based on the conclusions of Schiller et al. (2015). Instead, we relied on our analytical data and assessment of published data for EC002 and volcanic angrites.

The data for Asuka 881394 are not involved in the assessment of ^{26}Al distribution. This meteorite is more difficult to date with high precision than EC002 and volcanic angrites due to differences in mineralogy and chemical composition. Additionally, the ^{26}Al - ^{26}Mg in plagioclase in Asuka 881394 is disturbed (Wimpenny et al. 2019 and references therein), making determination of the initial abundance of ^{26}Al in this meteorite less precise and reliable.

We agree that "There are not a lot of samples that have been done to high fidelity," but our conclusions are based solely on the data for these rare, well-behaved samples.

Reviewer #3 (Remarks to the Author):

I think the authors have done a reasonable job of addressing my and the other reviewers' comments. I recommend publication.

Thank you.